# Asynchronous SGD Beats Minibatch SGD Under Arbitrary Delays

**Konstantin Mishchenko**       **Francis Bach**       **Mathieu Even**       **Blake Woodworth**

DI ENS, Ecole normale supérieure,
Université PSL, CNRS, INRIA
75005 Paris, France

## Abstract

The existing analysis of asynchronous stochastic gradient descent (SGD) degrades dramatically when any delay is large, giving the impression that performance depends primarily on the delay. On the contrary, we prove much better guarantees for the same asynchronous SGD algorithm regardless of the delays in the gradients, depending instead just on the number of parallel devices used to implement the algorithm. Our guarantees are strictly better than the existing analyses, and we also argue that asynchronous SGD outperforms synchronous minibatch SGD in the settings we consider. For our analysis, we introduce a novel recursion based on "virtual iterates" and delay-adaptive stepsizes, which allow us to derive state-of-the-art guarantees for both convex and non-convex objectives.

## 1 Introduction

We consider solving stochastic optimization problems of the form

$$\min_{\mathbf{x} \in \mathbb{R}^d} \{F(\mathbf{x}) := \mathbb{E}_{\xi \sim \mathcal{D}} f(\mathbf{x}; \xi)\}, \tag{1}$$

which includes machine learning (ML) training objectives, where $f(\mathbf{x}; \xi)$ represents the loss of a model parameterized by $\mathbf{x}$ on the datum $\xi$. Depending on the application, $\mathcal{D}$ could represent a finite dataset of size $n$ or a population distribution. In recent years, such stochastic optimization problems have continued to grow rapidly in size, both in terms of the dimension $d$ of the optimization variable—i.e., the number of model parameters in ML—and in terms of the quantity of data—i.e., the number of samples $\xi_1, \ldots, \xi_n \sim \mathcal{D}$ being used. With $d$ and $n$ regularly reaching the tens or hundreds of billions, it is increasingly necessary to use parallel optimization algorithms to handle the large scale and to benefit from data stored on different machines.

There are many ways of employing parallelism to solve (1), but the most popular approaches in practice are first-order methods based on stochastic gradient descent (SGD). At each iteration, SGD employs stochastic estimates of $\nabla F$ to update the parameters as $\mathbf{x}_k = \mathbf{x}_{k-1} - \gamma_k \nabla f(\mathbf{x}_{k-1}; \xi_{k-1})$ for an i.i.d. sample $\xi_{k-1} \sim \mathcal{D}$. Given $M$ machines capable of computing these stochastic gradient estimates $\nabla f(\mathbf{x}; \xi)$ in parallel, one approach to parallelizing SGD is what we call "Minibatch SGD." This refers to a synchronous, parallel algorithm that dispatches the current parameters $\mathbf{x}_{k-1}$ to each of the $M$ machines, waits while they compute and communicate back their gradient estimates $\mathbf{g}_{k-1}^1, \ldots, \mathbf{g}_{k-1}^M$, and then takes a minibatch SGD step $\mathbf{x}_k = \mathbf{x}_{k-1} - \gamma_k \cdot \frac{1}{M} \sum_{m=1}^M \mathbf{g}_{k-1}^m$. This is a natural idea with long history [16, 18, 55] and it is a commonly used in practice [e.g., 22]. However, since Minibatch SGD waits for all $M$ of the machines to finish computing their gradient estimates before updating, it proceeds only at the speed of the *slowest* machine.

There are several possible sources of delays: nodes may have heterogeneous hardware with different computational throughputs [23, 25], network latency can slow the communication of gradients, and

36th Conference on Neural Information Processing Systems (NeurIPS 2022).

nodes may even just drop out [45]. Slower "straggler" nodes can arise in many natural parallel settings including training ML models using multiple GPUs [14] or in the cloud, and sensitivity to these stragglers poses a serious problem for Minibatch SGD and other similar synchronous algorithms.

## 1.1 Asynchronous SGD

In this work, we consider a different, *asynchronous* parallel variant of SGD, which we define in Algorithm 1 and which has a long history [1, 3, 41]. For this method, whenever one of the $M$ machines finishes computing a stochastic gradient, the algorithm immediately uses it to take an SGD step, and then that machine begins computing a new stochastic gradient at the newly updated parameters. Because of the asynchronous updates, the other machines are now estimating the gradient at *out-of-date* parameters, so this algorithm ends up performing updates of the form[1]

$$\mathbf{x}_k = \mathbf{x}_{k-1} - \gamma_k \nabla f(\mathbf{x}_{k-\tau(k)}; \xi_{k-\tau(k)}),$$

where $\tau(k)$ is the "delay" of the gradient at iteration $k$, which is often much greater than one. Nevertheless, even though the updates are not necessarily well-aligned with the gradient of $F$ at the current parameters, the delays are usually not a huge problem in practice [17]. Asynchronous SGD has been particularly popular in reinforcement learning applications [36, 39] and federated learning [13, 43], providing significant speed-ups over Minibatch SGD.

However, the existing theoretical guarantees for Asynchronous SGD are disappointing, and the typical approach to analyzing the algorithm involves assuming that all of the delays are either the same, $\tau(k) = \tau$, or at least upper bounded, $\tau(k) \leqslant \tau_{\max}$ [1, 3, 29, 32, 48]. These analyses then show that the number of updates needed to reach accuracy $\epsilon$ grows linearly with $\tau_{\max}$, which could be very painful. Specifically, suppose we have two parallel workers—one fast device that needs just $1\mu$s to calculate a stochastic gradient, and one slow device that needs 1s. If we use these two machines to implement Asynchronous SGD, the delay of the slow device's gradients will be 1 million, because in the 1 second that we wait for the slow machine, the fast one will produce 1 million updates. Consequently, the analysis based on $\tau_{\max}$ degrades by a factor of 1 million. But on further reflection, Asynchronous SGD should actually do very well in this scenario, after all, 99.9999% of the SGD steps taken have gradients with no delay! Even if one update in a million has an enormous delay, it seems fairly clear that a few badly out-of-date gradients should not be enough to ruin the performance of SGD—a famously robust algorithm.

## 1.2 Speedup over Minibatch SGD

We will frequently compare Asynchronous SGD to Minibatch SGD (*e.g.*, Table 2), and to make this comparison easier, suppose for simplicity that each worker requires a fixed time of $s_m$ seconds per gradient computation, so in $S$ seconds, each machine computes $S/s_m$ stochastic gradients. Importantly, this translates into drastically different numbers of parameter updates for Asynchronous versus Minibatch SGD: the former takes one step per gradient computed, while the latter only takes one step for each gradient from the *slowest* machine. That is, Asynchronous and Minibatch SGD take

$$K_{\text{Async}} = \sum_{m=1}^{M} \frac{S}{s_m} \qquad \text{and} \qquad K_{\text{Mini}} = \min_{1 \leqslant m \leqslant M} \frac{S}{s_m} \tag{2}$$

total steps, respectively. So, it is easy to see that Asynchronous SGD takes *at least* $M$ times more steps than Minibatch SGD in any fixed amount of time, and even more than that when the machines have varying speeds. Soon, we will prove guarantees for Asynchronous SGD that match the guarantee for Minibatch SGD using exactly $M$ times fewer updates, meaning that our Asynchronous SGD guarantees are strictly better than the Minibatch SGD guarantees in terms of runtime.

## 1.3 Contributions and structure

In this work, we provide a new analysis for Asynchronous SGD, described in Section 2, which we use to prove better convergence guarantees. In contrast to the existing guarantees that are based

---

[1]Although this algorithm is asynchronous in the sense that different workers will have un-synchronized iterates, we nevertheless focus on a situation where each SGD step is an atomic/locked update of the parameters $\mathbf{x}_k$. This is in contrast to methods using lock-free updates, e.g., in the style of HOGWILD! [44], where different coordinates of the parameters might be updated and overwritten simultaneously by different workers.

Table 1: Comparison of the convergence rates for smooth objectives in terms of $K$, the total number of stochastic gradients used. For Minibatch SGD with $R$ updates with minibatch size $M$, it holds $K = MR$. For simplicity, we ignore all logarithmic and constant terms, including the coefficients in front of the exponents. The stated rates are upper bounds on $\mathbb{E}\left[\|\nabla F(\mathbf{x})\|^2\right]$ in the non-convex case, and $\mathbb{E}\left[F(\mathbf{x}) - F^*\right]$ in the (strongly) convex case.

| Method and reference | Convex | Strongly Convex | Non-Convex |
|---|---|---|---|
| Minibatch SGD[a]
Gower et al. [21]
Khaled and Richtárik [26] | $\dfrac{M}{K} + \dfrac{\sigma}{\sqrt{K}}$ | $e^{\frac{-\mu K}{LM}} + \dfrac{\sigma^2}{K}$ | $\dfrac{M}{K} + \dfrac{\sigma}{\sqrt{K}}$ |
| Asynchronous SGD
(fixed delay $\tau$)
Stich and Karimireddy [48] | $\dfrac{\tau}{K} + \dfrac{\sigma}{\sqrt{K}}$ | $e^{\frac{-\mu K}{L\tau}} + \dfrac{\sigma^2}{K}$ | $\dfrac{\tau}{K} + \dfrac{\sigma}{\sqrt{K}}$ |
| Asynchronous SGD
(arbitrary delays)
**Our work** | $\dfrac{M}{K} + \dfrac{\sigma}{\sqrt{K}}$ | $e^{\frac{-\mu K}{LM}} + \dfrac{\sigma^2}{K}$ | $\dfrac{M}{K} + \dfrac{\sigma}{\sqrt{K}}$ |

[a] Gower et al. [21] analyzed SGD in the strongly convex regime and Khaled and Richtárik [26] in the non-convex regime.

on $\tau_{\max}$, ours depend only on the number of workers, $M$, and show that Asynchronous SGD is better than the Minibatch SGD algorithm described earlier. For Lipschitz-continuous objectives, our results in Section 3 improve over existing Asynchronous and Minibatch SGD guarantees, and in the non-smooth, convex setting they are, in fact, minimax optimal. In Section 4, we prove state-of-the-art guarantees for smooth losses, which are summarized in Table 1. We do this by introducing a novel delay-adaptive stepsize schedule $\gamma_k \sim 1/\tau(k)$. The high-level intuition behind our proofs is that, although *some* of the gradients may have very large delay, *most* of the gradients have delay $\mathcal{O}(M)$, which is enough for good performance.

## 1.4 Related work

Asynchronous optimization has a long history. In the 1970s, Baudet [8] considered shared-memory asynchronous fixed-point iterations, and an early convergence result for Asynchronous SGD was established by Tsitsiklis et al. [49]. Recent analysis typically relies on bounded delays [1, 30, 44, 48]. Sra et al. [46] slightly relax this to random delays with bounded expectation. Zhou et al. [54] allowed delays to grow over time, but only show asymptotic convergence. Some algorithms try to adapt to the delays, but even these are not proven to perform well under arbitrary delays [35, 53]. For more examples of stochastic asynchronous algorithms, we refer readers to the surveys by Assran et al. [6], Ben-Nun and Hoefler [9].

In the online learning setting, Joulani et al. [24], McMahan and Streeter [33] studied an adaptive asynchronous SGD algorithm. Aviv et al. [7] proved better guarantees on bounded domains by introducing a projection, with rates depending on the average rather than maximum delay. However, their proof relies heavily on the assumption of a bounded domain, while ours applies for optimization over all of $\mathbb{R}^d$. Relatedly, Cohen et al. [15] prove guarantees for Asynchronous SGD that depend on the average delay, but their results only hold with probability $\frac{1}{2}$.

Most closely related to ours, Mania et al. [32] proposed and utilized the analysis tool of "virtual iterates" for Asynchronous SGD under bounded delays. Stich and Karimireddy [48] extended these results, albeit restricting delays to be constant, and Leblond et al. [29] considered lock-free updates. We use the same proof approach, but with a different virtual sequence and different, delay-adaptive stepsizes.

In a concurrent work, Koloskova et al. [28] used a similar technique to ours to study Asynchronous SGD with potentially unbounded delays. Their bounds are stated using empirical average of the delays rather than $M$. Compared to the work of Koloskova et al. [28], our theory includes guarantees for non-smooth problems as given in Theorem 1. They, on the other hand, have an extra result for the case of constant stepsize and bounded delays without assuming bounded gradients. Our Theorem 3 that covers non-convex, convex, and strongly convex problems has almost the same delay-adaptive

stepsize as their Theorem 8 that covers non-convex functions only. For heterogeneous data, we study standard Asynchronous SGD, whereas Koloskova et al. [28] used a special scheduling procedure to balance the workers, see also the comments after our Theorem 4.

More broadly, there are numerous other parallel training approaches that could improve upon Mini-batch SGD. Particularly popular lines of work include gradient compression [2, 10, 48], decentralized communication [31, 40], and local updates with infrequent communication [27, 34, 47, 50]. These are all orthogonal to asynchrony and can even be combined with it [see, e.g., 5, 19, 38].

### 1.5 Notation and problem setting

We consider solving the problem (1) under several standard [see, e.g., 11] combinations of conditions on the objective $F$. We denote the minimum of $F$ as $F^* := \min_{\mathbf{x}} F(\mathbf{x})$, an upper bound on the **initial suboptimality** $\Delta \geqslant F(\mathbf{x}_0) - F^*$, and an upper bound on the **initial distance** to the minimizer $B \geqslant \min\{\|\mathbf{x}_0 - \mathbf{x}^*\| : \mathbf{x}^* \in \arg\min_{\mathbf{x}} F(\mathbf{x})\}$. Here, and throughout this paper we focus on the Euclidean geometry and use $\|\cdot\|$ to denote the Euclidean norm; however, it is likely possible to extend our results to other geometries using similar arguments. Our optimization variable lies in $\mathbb{R}^d$, but our algorithm and results are dimension-free, meaning they hold for any $d$.

A function $F$ is $\mu$-**strongly convex** if for each $\mathbf{x}, \mathbf{y}$ and subgradient $\mathbf{g} \in \partial F(\mathbf{x})$, we have $F(\mathbf{y}) \geqslant F(\mathbf{x}) + \langle \mathbf{g}, \mathbf{y} - \mathbf{x} \rangle + \frac{\mu}{2}\|\mathbf{x} - \mathbf{y}\|^2$, and $F$ is **convex** if this holds for $\mu = 0$. When $F$ and $f$ are convex, we do not necessarily assume they are differentiable, but we abuse notation and use $\nabla F(\mathbf{x})$ and $\nabla f(\mathbf{x}; \xi)$ to denote an arbitrary subgradient at $\mathbf{x}$. The loss $f$ is $G$-**Lipschitz-continuous** if for each $\mathbf{x}, \mathbf{y}$ and $\xi$, we have $|f(\mathbf{x}; \xi) - f(\mathbf{y}; \xi)| \leqslant G\|\mathbf{x} - \mathbf{y}\|$, which also implies that for each $\mathbf{x}$, $\|\nabla f(\mathbf{x}; \xi)\| \leqslant G$ [11]. The objective $F$ is $L$-**smooth** if it is differentiable and its gradient is $L$-Lipschitz-continuous: for all $\mathbf{x}, \mathbf{y}$, $\|\nabla F(\mathbf{x}) - \nabla F(\mathbf{y})\| \leqslant L\|\mathbf{x} - \mathbf{y}\|$. We may also assume the stochastic gradients have $\sigma^2$-**bounded variance**, meaning that for all $\mathbf{x}$, $\mathbb{E}_{\xi \sim \mathcal{D}}\|\nabla f(\mathbf{x}; \xi) - \nabla F(\mathbf{x})\|^2 \leqslant \sigma^2$.

When the objective is (strongly) convex, we will obtain an upper bound on the expected suboptimality of our algorithm's output, $\tilde{\mathbf{x}}$, i.e., $\mathbb{E}F(\tilde{\mathbf{x}}) - F^* \leqslant \epsilon$ for some explicit $\epsilon$. On the other hand, when the objective is not convex, it is generally intractable to approximate global minima [42] so, as is common in the literature, we fall back to showing that the algorithm will find an approximate first-order stationary point of the objective: $\mathbb{E}\|\nabla F(\tilde{\mathbf{x}})\|^2 \leqslant \epsilon^2$.

Finally, we mainly focus on "homogeneous" optimization, where each machine computes each stochastic gradient using an i.i.d. sample $\xi \sim \mathcal{D}$. This is in contrast to the "heterogeneous" setting, where different machines have access to data drawn from different sources, meaning that stochastic gradients estimated on different machines can have different distributions (see Section 5).

**Delay notation.** The gradients used by Algorithm 1 may arrive out of order, so the parameters $\mathbf{x}_k$ at iteration $k$ will often be updated using stochastic gradients $\nabla f(\mathbf{x}_j; \xi_j)$ evaluated at out-of-date parameters $\mathbf{x}_j$ for $j < k - 1$; we therefore introduce additional notation for describing the delays. We use $m_k \in [M]$ to denote the index of the worker whose stochastic gradient estimate is used in iteration $k$ to compute $\mathbf{x}_k$. In addition, for each iteration $k$ and worker $m$, we introduce

$$\text{prev}(k, m) = \max\{j < k : m_j = m\} \qquad \text{and} \qquad \text{next}(k, m) = \min\{j \geqslant k : m_j = m\},$$

which denotes the index of the last iteration before $k$ in which machine $m$ returned a gradient, and the index of the first iteration after $k$ (inclusive) that machine $m$ will return a gradient, respectively. Accordingly, at iteration $k$, machine $m$ is in the process of estimating $\nabla f(\mathbf{x}_{\text{prev}(k,m)}; \xi_{\text{prev}(k,m)}^m)$. We define the current "delay" of this gradient as the number of iterations that have happened since $\text{prev}(k, m)$, i.e., $\tau(k, m) := k - \text{prev}(k, m)$. Abusing notation, we shorten to $\tau(k) := \tau(k, m_k)$ for the delay of the gradient used to compute $\mathbf{x}_k$.

## 2 Analysis of Asynchronous SGD via virtual iterates

The central idea in our analysis is to focus on a virtual iterate sequence, which tracks, roughly, how the parameters would have evolved if there were no delays. We note that this sequence is only used for the purpose of analysis, and is never actually computed. This technique is related to previous approaches [29, 32, 48], with the key difference being *which* virtual sequence we track. Specifically, in addition to $\mathbf{x}_0, \ldots, \mathbf{x}_K$—the actual sequence of iterates generated by Algorithm 1—we introduce

---
**Algorithm 1** Asynchronous SGD
---
1: **Input:** initialization $\mathbf{x}_0 \in \mathbb{R}^d$, stepsizes $\gamma_k > 0$
2: Each worker $m \in [M]$ begins calculating $\nabla f(\mathbf{x}_0; \xi_0^m)$
3: **for** $k = 1, 2, \ldots$ **do**
4:      Gradient $\nabla f(\mathbf{x}_{\text{prev}(k,m_k)}; \xi_{\text{prev}(k,m_k)}^{m_k})$ arrives from some worker $m_k$
5:      Update: $\mathbf{x}_k = \mathbf{x}_{k-1} - \gamma_k \nabla f(\mathbf{x}_{\text{prev}(k,m_k)}; \xi_{\text{prev}(k,m_k)}^{m_k})$
6:      Send $\mathbf{x}_k$ to worker $m_k$, which begins calculating $\nabla f(\mathbf{x}_k; \xi_k^{m_k})$
7: **end for**
---

the complementary sequence $\hat{\mathbf{x}}_1, \ldots, \hat{\mathbf{x}}_K$ which evolves according to

$$\hat{\mathbf{x}}_{k+1} = \hat{\mathbf{x}}_k - \hat{\gamma}_k \nabla f(\mathbf{x}_k; \xi_k^{m_k}),$$

$$\text{where} \quad \hat{\mathbf{x}}_1 := \mathbf{x}_0 - \sum_{m=1}^M \gamma_{\text{next}(1,m)} \nabla f(\mathbf{x}_0; \xi_0^m) \quad \text{and} \quad \hat{\gamma}_k := \gamma_{\text{next}(k+1,m_k)}. \tag{3}$$

This virtual sequence $\hat{\mathbf{x}}_{k+1}$ evolves *almost* according to SGD (without delays), although we note that it uses gradients evaluated at $\mathbf{x}_k$ rather than $\hat{\mathbf{x}}_k$. The stepsize used for this update, $\hat{\gamma}_k$, is the stepsize that is eventually used by Algorithm 1 when it takes a step using the gradient evaluated at $\mathbf{x}_k$. The core of our proofs is showing that $\mathbf{x}_k$ and $\hat{\mathbf{x}}_k$ remain close using the following Lemma:

**Lemma 1.** *Let $\{\mathbf{x}_k\}$ and $\{\hat{\mathbf{x}}_k\}$ be defined as in Algorithm 1 and* (3)*, respectively. Then for all $k \geqslant 1$*

$$\mathbf{x}_k - \hat{\mathbf{x}}_k = \sum_{m \in [M] \backslash \{m_k\}} \gamma_{\text{next}(k,m)} \nabla f(\mathbf{x}_{\text{prev}(k,m)}; \xi_{\text{prev}(k,m)}^m).$$

*Proof.* First, we expand the update of $\mathbf{x}_k$ in Algorithm 1, and of $\hat{\mathbf{x}}_k$ in (3). Denoting $\mathbf{e}_k = \mathbf{x}_k - \hat{\mathbf{x}}_k$,

$$\mathbf{e}_k = \mathbf{e}_{k-1} - \gamma_k \nabla f(\mathbf{x}_{\text{prev}(k,m_k)}; \xi_{\text{prev}(k,m_k)}^{m_k}) + \hat{\gamma}_{k-1} \nabla f(\mathbf{x}_{k-1}; \xi_{k-1}^{m_{k-1}})$$

$$= \mathbf{e}_1 - \sum_{j=2}^k \gamma_j \nabla f(\mathbf{x}_{\text{prev}(j,m_j)}; \xi_{\text{prev}(j,m_j)}^{m_j}) + \sum_{j=1}^{k-1} \hat{\gamma}_j \nabla f(\mathbf{x}_j; \xi_j^{m_j})$$

$$= \sum_{m=1}^M \gamma_{\text{next}(1,m)} \nabla f(\mathbf{x}_0; \xi_0^m) - \sum_{j=1}^k \gamma_j \nabla f(\mathbf{x}_{\text{prev}(j,m_j)}; \xi_{\text{prev}(j,m_j)}^{m_j}) + \sum_{j=1}^{k-1} \hat{\gamma}_j \nabla f(\mathbf{x}_j; \xi_j^{m_j}).$$

From here, we note that the second term, which comprises all of the gradients used by Algorithm 1 to make the first $k$ updates, can be rewritten as:

$$- \sum_{m=1}^M \gamma_{\text{next}(1,m)} \nabla f(\mathbf{x}_0; \xi_0^m) \mathbb{1}_{\{\text{next}(1,m) \leqslant k\}} - \sum_{i=1}^{k-1} \hat{\gamma}_i \nabla f(\mathbf{x}_i; \xi_i^{m_i}) \mathbb{1}_{\{\text{next}(i,m_i) \leqslant k\}},$$

and substituting into the expression for $\mathbf{e}_k$ above, there are $k$ cancellations and the claim follows. $\square$

How does this help us? For all of our results, our strategy is to show that the virtual iterates $\hat{\mathbf{x}}_k$ evolve essentially according to SGD (without delays), that $\|\mathbf{x}_k - \hat{\mathbf{x}}_k\|$ remains small throughout the algorithm's execution, and therefore that the Asynchronous SGD iterates $\mathbf{x}_k$ are nearly as good as SGD without delays. Lemma 1 is key for the second step. Whereas previous work tries to bound $\|\mathbf{x}_k - \hat{\mathbf{x}}_k\|$ by reasoning about the delays involved in the first $k$ updates, we observe that $\mathbf{x}_k - \hat{\mathbf{x}}_k$ is just the sum of $M - 1$ gradients, so our bound naturally incurs a dependence on $M$, but it is not directly affected by the delays themselves.

## 3 Convergence guarantees for Lipschitz losses

We begin by analyzing Algorithm 1 for convex, Lipschitz-continuous losses:

**Theorem 1.** *Let the objective $F$ be convex, let $f(\cdot; \xi)$ be $G$-Lipschitz-continuous for each $\xi$, and let there be a minimizer $\mathbf{x}^* \in \arg\min_{\mathbf{x}} F(\mathbf{x})$ for which $\|\mathbf{x}_0 - \mathbf{x}^*\| \leqslant B$. Then for any number of*

*iterations*[2] $K \geqslant M$, *Algorithm* 1 *with constant stepsize* $\gamma_k = \gamma = B/(G\sqrt{KM})$ *ensures*

$$\mathbb{E}\left[F\left(\frac{1}{K}\sum\nolimits_{k=1}^{K}\mathbf{x}_k\right) - F^*\right] \leqslant \frac{3GB\sqrt{M}}{\sqrt{K}}.$$

*Proof.* Let $\mathbf{x}^* \in \operatorname{argmin}_{\mathbf{x}} F(\mathbf{x})$ with $\|\mathbf{x}_0 - \mathbf{x}^*\| \leqslant B$. First, we follow the typical analysis of stochastic gradient descent [see, e.g., 11, Theorem 3.2] by expanding the update of $\hat{\mathbf{x}}_{k+1}$ from (3):

$$\begin{aligned}
\mathbb{E}\|\hat{\mathbf{x}}_{k+1} - \mathbf{x}^*\|^2 &= \mathbb{E}\big[\|\hat{\mathbf{x}}_k - \mathbf{x}^*\|^2 + \gamma^2\|\nabla f(\mathbf{x}_k;\xi_k^{m_k})\|^2 - 2\gamma\langle\nabla F(\mathbf{x}_k), \hat{\mathbf{x}}_k - \mathbf{x}^*\rangle\big] \\
&\leqslant \mathbb{E}\big[\|\hat{\mathbf{x}}_k - \mathbf{x}^*\|^2 + \gamma^2 G^2 - 2\gamma[F(\mathbf{x}_k) - F^*] + 2\gamma\langle\nabla F(\mathbf{x}_k), \mathbf{x}_k - \hat{\mathbf{x}}_k\rangle\big] \\
&\leqslant \mathbb{E}\big[\|\hat{\mathbf{x}}_k - \mathbf{x}^*\|^2 + \gamma^2 G^2 - 2\gamma[F(\mathbf{x}_k) - F^*] + 2\gamma G\|\mathbf{x}_k - \hat{\mathbf{x}}_k\|\big].
\end{aligned}$$

For the first inequality, we used the convexity of $F$ and that $f(\cdot;\xi)$ being $G$-Lipschitz implies $\|\nabla f(\mathbf{x};\xi)\| \leqslant G$ for all $\mathbf{x}$; for the second, we again used the $G$-Lipschitzness of $f(\cdot;\xi)$ along with the Cauchy-Schwarz inequality. Continuing as in the standard SGD analysis, we rearrange the expression, average over $K$, apply the convexity of $F$, and telescope the sum to conclude:

$$\mathbb{E}\left[F\left(\frac{1}{K}\sum_{k=1}^{K}\mathbf{x}_k\right) - F^*\right] \leqslant \mathbb{E}\left[\frac{\|\hat{\mathbf{x}}_1 - \mathbf{x}^*\|^2}{2\gamma K} + \frac{\gamma G^2}{2} + \frac{G}{K}\sum_{k=1}^{K}\|\mathbf{x}_k - \hat{\mathbf{x}}_k\|\right]. \tag{4}$$

The first two terms almost exactly match the guarantee of SGD with fixed stepsize $\gamma$. The main difference—and the place where Lemma 1 and the Lipschitzness of the losses plays a key role—is in bounding the third term. Since $\|\nabla f(\mathbf{x}_{\mathrm{prev}(k,m)};\xi_{\mathrm{prev}(k,m)}^m)\| \leqslant G$, we just use the triangle inequality:

$$\|\mathbf{x}_k - \hat{\mathbf{x}}_k\| = \left\|\sum_{m\in[M]\setminus\{m_k\}}\gamma\nabla f(\mathbf{x}_{\mathrm{prev}(k,m)};\xi_{\mathrm{prev}(k,m)}^m)\right\| \leqslant (M-1)\gamma G. \tag{5}$$

Combining this with $\|\hat{\mathbf{x}}_1 - \mathbf{x}^*\|^2 = \|\mathbf{x}_0 - \gamma\sum_{m=1}^{M}\nabla f(\mathbf{x}_0;\xi_0^m) - \mathbf{x}^*\|^2 \leqslant 2B^2 + 2\gamma^2 M^2 G^2$, and plugging in our stepsize in (4) completes the proof. $\qquad\square$

To understand this result, it is instructive to recall the worst-case performance of $K_{\mathrm{Mini}}$ steps of Minibatch SGD [42] in the setting of Theorem 1:

$$\mathbb{E}[F(\mathbf{x}_{\mathrm{Mini}}) - F^*] = \mathcal{O}\left(GB/\sqrt{K_{\mathrm{Mini}}}\right). \tag{6}$$

From this, we see that our guarantee for Asynchronous SGD in Theorem 1 matches the rate for $K_{\mathrm{Mini}} = K/M$ steps of Minibatch SGD. Furthermore, at least in the simplified model of Section 1.2, Asynchronous SGD takes at least $M$ times more steps than Minibatch SGD in a given span of time, and therefore Theorem 1 guarantees better performance than (6) in terms of runtime. Moreover, previous analyses of Asynchronous SGD [e.g., 48] provide guarantees with $\tau_{\max}$ replacing $M$ in our bound. Since necessarily $\tau_{\max} \geqslant M$, this means that our guarantee is never worse than the existing ones and it can be much better, for example, in a case where one severe straggler results in $\tau_{\max} \approx K$ but $M \ll K$. In fact, our guarantee in Theorem 1 is minimax optimal in the setting that we consider [see 51, Section 4.3], and as summarized in Table 2.

Finally, we emphasize that although $M$ appears in the numerator of the error guarantee in Theorem 1, this does *not* mean that the guarantee necessarily degrades when more parallel workers are added. In particular, adding more workers always means that more gradients will be calculated in any given amount of runtime. More concretely, in the model of Section 1.2 where the machines have fixed speeds, the expression for $K_{\mathrm{Async}} = K_{\mathrm{Async}}(S, s_1, \ldots, s_m)$ in (2) implies that adding an $(M+1)^{\mathrm{th}}$ machine gives a better guarantee whenever $s_{M+1}$ is smaller than the harmonic mean of $s_1, \ldots, s_M$:

$$s_{M+1} \leqslant \left(\frac{1}{M}\sum_{m=1}^{M}\frac{1}{s_m}\right)^{-1} \implies \frac{M+1}{K_{\mathrm{Async}}(S, s_1, \ldots, s_{M+1})} \leqslant \frac{M}{K_{\mathrm{Async}}(S, s_1, \ldots, s_M)}.$$

Following the same high-level approach, we can also analyze Algorithm 1 for non-convex objectives, the proof being deferred to Appendix C.

---

[2] W.l.o.g. we can take $M \leqslant K$ because at most $K$ of the workers are actually able participate in the first $K$ updates of Algorithm 1, and machines that do not participate can simply be ignored in the analysis.

Table 2: We compare optimization terms in the smooth and convex setting and the resulting speed-ups in the fixed-computation-speed model of Section 1.2. We denote $s_{\max} := \max_m s_m$ and approximate the maximum delay as $\tau_{\max} = \sum_{m=1}^{M} s_{\max}/s_m$. The speedup is the largest factor $\alpha$ such that Asynchronous SGD attains the same error in $S$ seconds as Minibatch SGD would in $\alpha S$ seconds.

| Method | # of Updates | Optimization Term | Speedup |
|---|---|---|---|
| **Minibatch SGD** | $R = \frac{S}{s_{\max}}$ | $\mathcal{O}\left(\frac{1}{R}\right)$ | 1 |
| **Asynchronous SGD** (prior works) | $K = \sum_{m=1}^{M} \frac{S}{s_m}$ | $\mathcal{O}\left(\frac{\tau_{\max}}{K}\right)$ | 1 |
| **Asynchronous SGD** (our work) | $K = \sum_{m=1}^{M} \frac{S}{s_m}$ | $\mathcal{O}\left(\frac{M}{K}\right)$ | $\frac{1}{M} \sum_{m=1}^{M} \frac{s_{\max}}{s_m} \geqslant 1$ |

**Theorem 2.** *Let $F$ be $L$-smooth, $f(\cdot;\xi)$ be $G$-Lipschitz-continuous for every $\xi$, let $\Delta \geqslant F(\mathbf{x}_0) - F^*$, and let the stochastic gradients have variance at most $\sigma^2$. Then Algorithm 1 with stepsize $\gamma = \min\left\{\frac{1}{2ML}, \sqrt{\frac{\Delta}{L\sigma^2 K}}, \left(\frac{\Delta}{L^2 M^2 G^2 K}\right)^{1/3}\right\}$, ensures for any $K \geqslant M$*

$$\frac{1}{K} \sum_{k=1}^{K} \mathbb{E}\|\nabla F(\mathbf{x}_k)\|^2 = \mathcal{O}\left(\frac{ML\Delta}{K} + \sqrt{\frac{L\Delta\sigma^2}{K}} + \left(\frac{ML\Delta G}{K}\right)^{2/3}\right).$$

## 4 Convergence guarantees for non-Lipschitz losses

Now, we analyze smooth but not necessarily Lipschitz-continuous losses. The previous proofs relied crucially on the gradients being bounded in norm in order to control $\mathbf{x}_k - \hat{\mathbf{x}}_k$. For general smooth losses, the situation is more difficult because $\|\nabla f(\mathbf{x}_{\text{prev}(k,m)}; \xi_{\text{prev}(k,m)}^m)\|$ could be large. Our solution to this issue is to introduce a new *delay-adaptive* stepsize schedule $\gamma_k \sim 1/\tau(k)$, which we show allows for sufficient control over $\|\mathbf{x}_k - \hat{\mathbf{x}}_k\|$. Similar stepsizes have been considered by Wu et al. [52] for the PIAG algorithm, while the stepsizes for Asynchronous SGD used in previous analyses typically scale with $1/\tau_{\max}$ [e.g., 48]. Thus, our analysis shows that we can get away with a more aggressive stepsize to get better rates. However, our stepsize choice could be problematic if it were correlated with the noise in the stochastic gradients because our proofs involve the step:

$$\mathbb{E}_{\xi_{\text{prev}(k,m_k)}^{m_k} \sim \mathcal{D}}\left[\gamma_k \nabla f(\mathbf{x}_{\text{prev}(k,m_k)}; \xi_{\text{prev}(k,m_k)}^{m_k})\right] = \gamma_k \nabla F(\mathbf{x}_{\text{prev}(k,m_k)}).$$

Therefore, we introduce the following assumption about the relationship between the delays and data:

**Assumption 1.** *The stochastic sequences $(\xi_1, \xi_2, \ldots)$ and $(\tau(0), \tau(1), \ldots)$ are independent.*

This assumption holds, for example, when it takes a fixed amount of computation to evaluate any stochastic gradient, and the combination of the (potentially heterogeneous) computational throughput on the different workers and network latency gives rise to the delays. However, this can fail to hold, for instance, when training a model with variable-length inputs, in which case the delays will probably depend on the length of the input sequences, so it will likely also be related to the gradient noise. The next Theorem shows our convergence guarantees under Assumption 1 for Asynchronous SGD with novel delay-adaptive stepsizes scaling with $1/\tau(k)$:

**Theorem 3.** *Suppose $F$ is $L$-smooth, that Assumption 1 holds, that $B \geqslant \mathbb{E}\|\mathbf{x}_0 - \mathbf{x}^*\|^2$ for some minimizer $\mathbf{x}^*$, and $\Delta \geqslant \mathbb{E}F(\mathbf{x}_0) - F^*$. Then there exist numerical constants $c_1, c_2, c_3, c_4$ such that:*

    *1. For convex $F$, $K \geqslant M$ and $\gamma_k = \min\left\{\frac{1}{4L\tau(k)}, \frac{1}{4ML}, \frac{B}{\sigma\sqrt{K}}\right\}$ Algorithm 1 ensures*

$$\mathbb{E}\left[F(\tilde{\mathbf{x}}_K) - F^*\right] \leqslant c_1 \cdot \left(\frac{MLB^2}{K} + \frac{\sigma B}{\sqrt{K}}\right),$$

    *where $\tilde{x}_K$ is the weighted average[3] of $\mathbf{x}_1, \ldots, \mathbf{x}_K$ defined in (20).*

---

[3] The weight on each iterate $\mathbf{x}_k$ is proportional to $\hat{\gamma}_k$, which depends on the eventual delay of $\nabla f(\mathbf{x}_k; \xi_k^{m_k})$, and is thus not yet known at iteration $k$. However, on line 4 of Algorithm 1, the worker could simply return both its gradient and the point at which it was evaluated, so that the term $\hat{\gamma}_k \mathbf{x}_k$ can simply be added at iteration $\text{next}(k+1, m_k)$ rather than at iteration $k$, so there is not need to store all of the previous iterates.

2. *For $\mu$-strongly convex $F$, $K \geqslant 3M$, and $\gamma_k = \min\Big\{\frac{\exp\big(\frac{-\mu\tau(k)}{4ML}\big)}{4L\tau(k)}, \frac{1}{8ML}, \frac{504\ln\big(e+\frac{\mu^2 K^2 B^2}{\sigma^2}\big)}{\mu K}\Big\}$*
   *Algorithm 1 ensures*

$$\mathbb{E}[F(\tilde{\mathbf{x}}_K) - F^*] \leqslant c_2 \cdot \left( MLB^2 \exp\left(-\frac{c_3 K\mu}{ML}\right) + \frac{\sigma^2}{\mu K}\log\left(e + \frac{\mu^2 K^2 B^2}{\sigma^2}\right)\right),$$

   *where $\tilde{\mathbf{x}}_K$ is the weighted average of $\mathbf{x}_1, \ldots, \mathbf{x}_K$ defined in (22).*

3. *For non-convex $F$, $K \geqslant M$, and $\gamma_k = \min\Big\{\frac{1}{4L\tau(k)}, \frac{1}{2ML}, \sqrt{\frac{\Delta}{KL\sigma^2}}\Big\}$ Algorithm 1 ensures*

$$\mathbb{E}\big[\|\nabla F(\tilde{\mathbf{x}}_K)\|^2\big] \leqslant c_4 \cdot \left(\frac{ML\Delta}{K} + \sqrt{\frac{L\Delta\sigma^2}{K}}\right),$$

   *where $\tilde{\mathbf{x}}_K$ is randomly chosen from $\mathbf{x}_1, \ldots, \mathbf{x}_K$ according to (25).*

We defer the complete proof of the three parts to Appendices D.1, D.2, and D.3. However, to give a sense of our approach, we will now sketch the ideas behind the proof of Theorem 3.1:

*Proof sketch.* We begin following the standard analysis of SGD for smooth, convex objectives. We expand the update $\hat{\mathbf{x}}_{k+1}$ from (3), and use the $L$-smoothness of the objective and the fact that the stepsizes are less than $1/(4L)$ to obtain the inequality

$$\mathbb{E}\|\hat{\mathbf{x}}_{k+1} - \mathbf{x}^*\|^2 \leqslant \mathbb{E}\left[\|\hat{\mathbf{x}}_k - \mathbf{x}^*\|^2 - \frac{3}{2}\hat{\gamma}_k(F(\mathbf{x}_k) - F^*) + \hat{\gamma}_k^2\sigma^2 + 2\hat{\gamma}_k\left\langle \nabla f(\mathbf{x}_k; \xi_k^{m_k}), \mathbf{x}_k - \hat{\mathbf{x}}_k\right\rangle\right].$$

A key step above uses Assumption 1 to show $\hat{\gamma}_k$ is independent of the gradient noise, meaning

$$\mathbb{E}[\hat{\gamma}_k\left\langle \nabla f(\mathbf{x}_k; \xi_k^{m_k}), \mathbf{x}_k - \mathbf{x}^*\right\rangle] = \mathbb{E}[\hat{\gamma}_k\left\langle \nabla F(\mathbf{x}_k), \mathbf{x}_k - \mathbf{x}^*\right\rangle] \geqslant \mathbb{E}[\hat{\gamma}_k(F(\mathbf{x}_k) - F^*)].$$

This is, so far, essentially identical to the first step of the typical SGD analysis, the only difference being the fourth term. Rearranging the expression and summing over $k$, this implies

$$\frac{3}{2}\sum_{k=1}^K \mathbb{E}[\hat{\gamma}_k(F(\mathbf{x}_k) - F^*)] \leqslant \mathbb{E}\left[\|\hat{\mathbf{x}}_1 - \mathbf{x}^*\|^2 + \sigma^2\sum_{k=1}^K \hat{\gamma}_k^2 + 2\sum_{k=1}^K \hat{\gamma}_k\left\langle \nabla f(\mathbf{x}_k; \xi_k^{m_k}), \mathbf{x}_k - \hat{\mathbf{x}}_k\right\rangle\right]. \quad (7)$$

This still resembles the typical SGD analysis except for the third term on the right hand side, but we begin to see divergence here. First, we must bound this third term, for which we introduce the following Lemma, proven by using Lemma 1 to write $\mathbf{x}_k - \hat{\mathbf{x}}_k$ as a sum of gradients:

**Lemma 2.** *In the setting of Theorem 3.1,*

$$\mathbb{E}\left[2\sum_{k=1}^K \hat{\gamma}_k\left\langle \nabla f(\mathbf{x}_k; \xi_k^{m_k}), \mathbf{x}_k - \hat{\mathbf{x}}_k\right\rangle\right] \leqslant \mathbb{E}\left[\frac{B^2}{16} + \sum_{k=1}^K \hat{\gamma}_k(F(\mathbf{x}_k) - F^*)\right].$$

Combining this with (7) and using the convexity of $F$, we further conclude that

$$\mathbb{E}\left[F\left(\frac{\sum_{k=1}^K \hat{\gamma}_k\mathbf{x}_k}{\sum_{k=1}^K \hat{\gamma}_k}\right) - F^*\right] \leqslant \mathbb{E}\left[\frac{2}{\sum_{k=1}^K \hat{\gamma}_k}\left(\frac{B^2}{16} + \|\hat{\mathbf{x}}_1 - \mathbf{x}^*\|^2 + \sigma^2\sum_{k=1}^K \hat{\gamma}_k^2\right)\right]. \quad (8)$$

This shows that a weighted average of the iterates—with $\mathbf{x}_k$ unconventionally weighted by the stepsize $\hat{\gamma}_k$ rather than typical uniform weights—attains suboptimality inversely proportional to the sum of the stepsizes. To conclude, we lower bound the sum of stepsizes in the denominator by a term scaling with $\min\{K/ML, B\sqrt{K}/\sigma\}$. All remaining proof details can be found in Appendix D. $\quad\square$

To understand the implications, we recall the guarantees for $R$ steps of Minibatch SGD using minibatches of size $M$ in the setting of Theorem 3, which are (ignoring all constants) [20, 42]: $\mathbb{E}[F(\mathbf{x}_{\text{Mini}}) - F^*] \leqslant \frac{LB^2}{R} + \frac{\sigma B}{\sqrt{RM}}$ in the convex setting, $\mathbb{E}[F(\mathbf{x}_{\text{Mini}}) - F^*] \leqslant LB^2\exp(-\mu R/L) + \frac{\sigma^2}{\mu RM}$ in the strongly convex setting, and $\mathbb{E}\|\nabla F(\mathbf{x}_{\text{Mini}})\|^2 \leqslant \frac{L\Delta}{R} + \sqrt{L\Delta\sigma^2/RM}$ in the non-convex setting. Comparing these to the guarantees for Asynchronous SGD in Theorem 3.1–3, we see that up to constants (and logarithmic factors in the strongly convex case), our results match the guarantees of $R = K/M$ steps of Minibatch SGD with minibatch size $M$. As discussed in Section 1.2, each

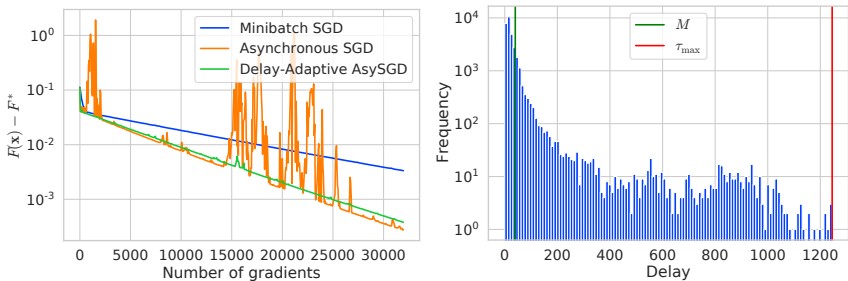

Figure 1: We ran an experiment on a simple least-squares problem with random data and tuned all stepsizes. On the left, we see that after a fixed number of gradients are used, Asynchronous SGD is slightly better than Minibatch SGD but it is also quite unstable, whereas with delay-adaptive stepsizes it is both fast and stable. On the right, we can see the distribution of the delays (note that y-axis is in log scale). Additional details about the experiment can be found in Appendix A.

update of Minibatch SGD takes the time needed by the *slowest* machine, meaning that in any given amount of time, Asynchronous SGD will complete at least $M$ times more updates than Minibatch would. Therefore, the guarantees in Theorem 3 imply *strictly better* performance for Algorithm 1 than for Minibatch SGD in terms of guaranteed error after a fixed amount of time, as illustrated in Table 2. Figure 1 depicts a simple experiment demonstrating this phenomenon in practice.

The first "optimization" terms in our guarantees match $K/M$ steps of exact gradient descent, completely irrespective of the delays. In the convex cases, it is likely that these could be "accelerated" to scale with $(K/M)^{-2}$ or $\exp(-K\sqrt{\mu}/M\sqrt{L})$, but at the expense of a more complex algorithm and analysis. However, by analogy to existing lower bounds [12, 51] we conjecture that the optimization terms in Theorem 3.1–2 are the best "unaccelerated" rate we could hope for, and that the optimization term in Theorem 3.3 is optimal. Moreover, despite the delays, the second "statistical" terms in our guarantees are minimax optimal amongst all algorithms that use $K$ stochastic gradients [4, 42]. So, when the statistical terms dominate the rate, our guarantees are unimprovable in the worst case, and in fact they even match what could be guaranteed by $K$ steps of SGD without delays. Furthermore, since the optimization terms decrease faster with $K$, in a realistic scenario where $K \gg M$, the statistical term will eventually dominate the rate and our algorithm will have optimal performance with no penalty from the delayed gradients at all.

## 5 Heterogeneous data setting

Finally, we extend our results to the heterogeneous data setting, where each worker $m$ possesses its own local data distribution $\mathcal{D}_m$, giving rise to a local objective $F_m$. So the optimization problem in the heterogeneous setting is:

$$\min_{\mathbf{x}\in\mathbb{R}^d}\left\{F(x) = \frac{1}{M}\sum_{m=1}^{M}F_m(\mathbf{x})\right\}, \quad \text{where} \quad F_m(\mathbf{x}) = \mathbb{E}_{\xi\sim\mathcal{D}_m}[f_m(\mathbf{x};\xi)]. \tag{9}$$

In this setting, we define Asynchronous SGD the same way, with worker $m$ having access to the stochastic gradients $\nabla f_m(\cdot;\xi)$ and updates taking the form:

$$\mathbf{x}_k = \mathbf{x}_{k-1} - \gamma_k \nabla f_{m_k}\big(\mathbf{x}_{\text{prev}(k,m_k)}; \xi_{\text{prev}(k,m_k)}^{m_k}\big). \tag{10}$$

It is generally impossible to show that Asynchronous SGD will work well in this setting. Specifically, if $F_2 = \cdots = F_M = 0$, then the time needed to optimize $F$ is entirely dependent on the speed of the first worker. Moreover, with arbitrary delays, we might only get a single gradient update from the first worker, leaving us no hope of any useful guarantee. To avoid this issue and to be able to apply our analysis, we will require that the gradient dissimilarities between the local functions are bounded and that the identity of the worker used in iteration $k$ is independent of the gradient noise:

**Assumption 2.** *There exists $\zeta \geqslant 0$ such that $\|\nabla F_m(\mathbf{x}) - \nabla F(\mathbf{x})\|^2 \leqslant \zeta^2$ for all $m$ and $\mathbf{x} \in \mathbb{R}^d$, $\mathbb{E}[\mathbf{g}_k|\mathbf{x}_k, m_k] = \nabla F_{m_k}(\mathbf{x}_k)$, and $\mathbb{E}[\|\mathbf{g}_k - \nabla F_{m_k}(\mathbf{x}_k)\|^2|\mathbf{x}_k, m_k] \leqslant \sigma^2$.*

Equipped with the new assumption, we can provide an analogue of Theorem 3 for problem (9):

**Theorem 4.** *In the setting of Theorem 3 with the addition of Assumption 2, for $\gamma_k = \min\{\frac{1}{8L\tau(k)}, \frac{1}{4ML}, \sqrt{\Delta/(KL\sigma^2)}\}$, the updates described in* (10) *ensure for a numerical constant $c$:*

$$\mathbb{E}\big[\|\nabla F(\tilde{\mathbf{x}}_K)\|^2\big] \leqslant c \cdot \left(\frac{ML\Delta}{K} + \sqrt{\frac{L\Delta\sigma^2}{K}} + \zeta^2\right),$$

*where $\tilde{\mathbf{x}}_k$ is randomly chosen from $\mathbf{x}_1, \ldots, \mathbf{x}_K$ according to* (28).

Comparing Theorem 4—which we prove in Appendix E—with Theorem 3.3, we see that the exact same rate is achieved in the heterogeneous setting up to the additive term $\zeta^2$. As described above, we cannot really expect good performance for arbitrarily heterogeneous losses under arbitrary delays, so some dependence on $\zeta^2$ is unavoidable. Moreover, in many natural settings, $\zeta^2$ can be quite small. For instance, when heterogeneity arises because a large i.i.d. dataset is partitioned across the $M$ machines, the degree of heterogeneity $\zeta^2$ will be inversely proportional to the number of samples assigned to each worker. So although Asynchronous SGD is not well-suited to problem (9) when the delays can be arbitrarily large, at least under Assumption 2, it does not totally break down or diverge.

While Assumption 2 used in Theorem 4 seems unavoidable for studying Asynchronous SGD, it can be circumvented when studying other asynchronous methods. In particular, Koloskova et al. [28] proposed to sample a random worker at each iteration $k$ and add the current points $\mathbf{x}_k$ to its list of points for computing the gradient. Under an extra assumption on the delay pattern, Koloskova et al. [28] proved convergence of this method to a point with zero gradient, without extra errors. Unfortunately, unlike Asynchronous SGD, their method would not have the speedup shown in Table 2, since after a large number of iterations $K$, all workers would be expected to compute roughly $\frac{K}{M}$ gradients. Since this puts the same load on the slowest worker as Minibatch SGD, the algorithm proposed by Koloskova et al. [28] is useful only when there is no worker that is fundamentally slower than the others. Our theory, in turn, has an extra $\zeta^2$ error term, but does not require waiting for the slow workers.

## Conclusion

This work studies Asynchronous SGD via a virtual-iterate analysis: we prove that in a vast variety of settings – convex/Lipschitz, non-convex/Lipschitz/smooth, convex/smooth, strongly-convex/smooth and non-convex/smooth – the convergence guarantees of Asynchronous SGD improve over that of Minibatch SGD, *irrespectively of the delay sequence*. We obtained these results by leveraging delay-adaptive stepsizes and proving that Asynchronous SGD is only slowed down by the number of stochastic gradients being computed, rather than the longest delay. We also extend one of these results to the case of heterogeneous data and show that Asynchronous SGD can converge to an approximate stationary point with the error controlled by a data dissimilarity constant.

## Acknowledgements

This work was supported by the French government under the management of the Agence Nationale de la Recherche as part of the "Investissements d'avenir" program, reference ANR-19-P3IA-0001 (PRAIRIE 3IA Institute). We also acknowledge support from the European Research Council (grant SEQUOIA 724063).

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
