# A   Experimental details

To showcase how our theory matches the numerical performance of Asynchronous SGD, we run experiments on a simple quadratic objective with random data. We run our experiments on a single node with 48 logical cores and set $M = 40$. We use the Ray package [37] to parallelize the execution and follow the official documentation for the implementation[4] of asynchronous training. All delays appearing in the runs are not simulated and come from the execution on CPUs. For further reproducibility, we released our code online[5].

# B   Intermediate results

**Lemma 3.** *Under Assumption [1] and the $\sigma^2$ variance bound, if $\gamma_k$ depends only on $k$ and $\tau(k)$ for each $k$, then for all $k \geqslant 1$,*

$$\mathbb{E}\|\mathbf{x}_k - \hat{\mathbf{x}}_k\|^2 \leqslant 2\mathbb{E}\left[\sum_{m \in [M] \setminus \{m_k\}} \gamma^2_{\mathrm{next}(k,m)}\Big[\sigma^2 + (M-1)\big\|\nabla F(\mathbf{x}_{\mathrm{prev}(k,m)})\big\|^2\Big]\right].$$

*Proof.* By Lemma [1], we have

$$\mathbb{E}\|\mathbf{x}_k - \hat{\mathbf{x}}_k\|^2 = \mathbb{E}\left\|\sum_{m \in [M] \setminus \{m_k\}} \gamma_{\mathrm{next}(k,m)} \nabla f(\mathbf{x}_{\mathrm{prev}(k,m)}; \xi^m_{\mathrm{prev}(k,m)})\right\|^2$$

$$\leqslant 2\mathbb{E}\left\|\sum_{m \in [M] \setminus \{m_k\}} \gamma_{\mathrm{next}(k,m)} \nabla F(\mathbf{x}_{\mathrm{prev}(k,m)})\right\|^2$$

$$+ 2\mathbb{E}\left\|\sum_{m \in [M] \setminus \{m_k\}} \gamma_{\mathrm{next}(k,m)}\big(\nabla f(\mathbf{x}_{\mathrm{prev}(k,m)}; \xi^m_{\mathrm{prev}(k,m)}) - \nabla F(\mathbf{x}_{\mathrm{prev}(k,m)})\big)\right\|^2.$$

Notice that we had to use Young's inequality to separate expectations from the variance terms since the variance in the vectors is not independent. The first term can be bounded using Young's inequality as follows,

$$\mathbb{E}\left\|\sum_{m \in [M] \setminus \{m_k\}} \gamma_{\mathrm{next}(k,m)} \nabla F(\mathbf{x}_{\mathrm{prev}(k,m)})\right\|^2 \leqslant \sum_{m \in [M]} \gamma^2_{\mathrm{next}(k,m)}\|\nabla F(\mathbf{x}_{\mathrm{prev}(k,m)})\|^2.$$

To bound the second term, assume without loss of generality that $\mathrm{prev}(k,1) < \mathrm{prev}(k,2) < \cdots < \mathrm{prev}(k,M)$. In addition, denote

$$\boldsymbol{\theta}_m = \gamma_{\mathrm{next}(k,m)}\big(\nabla f(\mathbf{x}_{\mathrm{prev}(k,m)}; \xi^m_{\mathrm{prev}(k,m)}) - \nabla F(\mathbf{x}_{\mathrm{prev}(k,m)})\big).$$

Then, we have for any $m$

$$\mathbb{E}\left[\|\boldsymbol{\theta}_m\|^2\right] = \gamma^2_{\mathrm{next}(k,m)}\mathbb{E}\|\nabla f(\mathbf{x}_{\mathrm{prev}(k,m)}; \xi^m_{\mathrm{prev}(k,m)}) - \nabla F(\mathbf{x}_{\mathrm{prev}(k,m)})\|^2 \leqslant \gamma^2_{\mathrm{next}(k,m)}\sigma^2.$$

Moreover, for any $m \in [1, M-1]$, the stochastic gradient of worker $m+1$ has conditional expectation

$$\mathbb{E}\left[\nabla f(\mathbf{x}_{\mathrm{prev}(k,m+1)}; \xi^{m+1}_{\mathrm{prev}(k,m+1)}) \mid \nabla f(\mathbf{x}_{\mathrm{prev}(k,m)}; \xi^m_{\mathrm{prev}(k,m)})\right] = \nabla F(\mathbf{x}_{\mathrm{prev}(k,m+1)}),$$

---

[4]https://docs.ray.io/en/latest/ray-core/examples/plot_parameter_server.html#asynchronous-parameter-server-training
[5]https://github.com/konstmish/asynchronous_sgd

so $\mathbb{E}\left[\boldsymbol{\theta}_{m+1} \mid \boldsymbol{\theta}_1, \ldots, \boldsymbol{\theta}_m\right] = 0$. This allows us to obtain by induction,

$$
\begin{aligned}
\mathbb{E}\left[\|\sum_{j=1}^{m+1}\boldsymbol{\theta}_j\|^2\right] &= \mathbb{E}\left[\|\sum_{j=1}^{m}\boldsymbol{\theta}_j\|^2 + 2\langle\sum_{j=1}^{m}\boldsymbol{\theta}_j, \boldsymbol{\theta}_{m+1}\rangle + \|\boldsymbol{\theta}_{m+1}\|^2\right] \\
&\leqslant \mathbb{E}\left[\|\sum_{j=1}^{m}\boldsymbol{\theta}_j\|^2 + 2\langle\sum_{j=1}^{m}\boldsymbol{\theta}_j, \boldsymbol{\theta}_{m+1}\rangle\right] + \gamma_{\text{next}(k,m+1)}^2\sigma^2 \\
&\leqslant \mathbb{E}\left[\sum_{j=1}^{m}\hat{\gamma}_{t-\tau_j}^2\sigma^2 + 2\langle\sum_{j=1}^{m}\boldsymbol{\theta}_j, \boldsymbol{\theta}_{m+1}\rangle\right] + \gamma_{\text{next}(k,m+1)}^2\sigma^2 \\
&= \mathbb{E}\left[2\langle\sum_{j=1}^{m}\boldsymbol{\theta}_j, \boldsymbol{\theta}_{m+1}\rangle\right] + \sum_{j=1}^{m+1}\gamma_{\text{next}(k,j)}^2\sigma^2.
\end{aligned}
$$

The remaining scalar product is, in fact, equal to zero. Indeed, by the tower property of expectation,

$$
\begin{aligned}
\mathbb{E}\left[\langle\sum_{j=1}^{m}\boldsymbol{\theta}_j, \boldsymbol{\theta}_{m+1}\rangle\right] &= \mathbb{E}\left[\mathbb{E}\left[\langle\sum_{j=1}^{m}\boldsymbol{\theta}_j, \boldsymbol{\theta}_{m+1}\rangle \mid \boldsymbol{\theta}_1, \ldots, \boldsymbol{\theta}_m\right]\right] \\
&= \mathbb{E}\left[\langle\sum_{j=1}^{m}\boldsymbol{\theta}_j, \mathbb{E}\left[\boldsymbol{\theta}_{m+1} \mid \boldsymbol{\theta}_1, \ldots, \boldsymbol{\theta}_m\right]\rangle\right] \\
&= 0.
\end{aligned}
$$

Therefore,

$$
\mathbb{E}\left\|\sum_{m\in[M]\setminus\{m_k\}}\gamma_{\text{next}(k,m)}(\nabla f(\mathbf{x}_{\text{prev}(k,m)}; \xi_{\text{prev}(k,m)}^m) - \nabla F(\mathbf{x}_{\text{prev}(k,m)}))\right\|^2
$$
$$
\leqslant \sum_{m\in[M]\setminus\{m_k\}}\gamma_{\text{next}(k,m)}^2\sigma^2.
$$

$\square$

Lemma 3 is very useful whenever stochastic gradients are not guaranteed to be bounded. If they were bounded, we could immediately show that $\mathbb{E}\|\mathbf{x}_k - \hat{\mathbf{x}}_k\|^2$ is small regardless of the delays, as was done in the proof of Theorem 1, see equation (5). For non-Lipschitz losses, however, $\mathbb{E}\|\mathbf{x}_k - \hat{\mathbf{x}}_k\|^2$ is not guaranteed to be finite, and Lemma 3 is required to show that $\hat{\mathbf{x}}_k$ and $\mathbf{x}_k$ stay sufficiently close to each other.

### B.1 A caveat in the proof of Lemma 3

In the proof above, our first step was to split conditional expectations from the conditional variances using inequality $\|a + b\|^2 \leqslant 2\|a\|^2 + 2\|b\|^2$. One may wonder why we did not use instead the decomposition $\mathbb{E}\|X\|^2 = \|\mathbb{E}X\|^2 + \mathbb{E}\|X - \mathbb{E}X\|^2$ that holds for any random variable $X$ with finite variance. While this identity is simpler, it also leads to full expectations inside all norms, producing $\gamma_{\text{next}(k,m)}(\nabla f(\mathbf{x}_{\text{prev}(k,m)}; \xi_{\text{prev}(k,m)}^m) - \mathbb{E}\left[\nabla F(\mathbf{x}_{\text{prev}(k,m)})\right])$. Our proof, on the other hand, relied on the identity

$$
\mathbb{E}\left[\langle\boldsymbol{\theta}_j, \boldsymbol{\theta}_{m+1}\rangle\right] = \mathbb{E}\left[\langle\boldsymbol{\theta}_j, \mathbb{E}\left[\boldsymbol{\theta}_{m+1} \mid \boldsymbol{\theta}_1, \ldots, \boldsymbol{\theta}_m\right]\rangle\right] = 0
$$

for all $j \leqslant m$. Notice that the same would not be true if we defined $\boldsymbol{\theta}_{m+1}$ with $\mathbb{E}\left[\nabla F(\mathbf{x}_{\text{prev}(k,m)})\right]$ instead of $\nabla F(\mathbf{x}_{\text{prev}(k,m)})$, since in general $\mathbb{E}\left[\nabla f(\mathbf{x}_{\text{prev}(k,m)}; \xi_{\text{prev}(k,m)}^m) \mid \boldsymbol{\theta}_1, \ldots, \boldsymbol{\theta}_m\right] = \nabla F(\mathbf{x}_{\text{prev}(k,m)}) \neq \mathbb{E}\left[\nabla F(\mathbf{x}_{\text{prev}(k,m)})\right]$.

### B.2 A lemma for products with delays

The next lemma is a general statement about sequences with delays, which we will use to bound various error terms in our proofs.

**Lemma 4.** *For any positive sequences $\{a_k\}$, $\{b_k\}$, and $\{c_k\}$, it holds*

$$\sum_{k=1}^{K} \sum_{m \in [M] \setminus \{m_k\}} a_k b_{\text{prev}(k,m)} c_{\text{next}(k,m)} = b_0 \sum_{m=1}^{M} c_{\text{next}(1,m)} \sum_{j=1}^{\min\{\text{next}(1,m)-1, K\}} a_j$$

$$+ \sum_{k=1}^{K-1} b_k c_{\text{next}(k+1,m_k)} \sum_{j=k+1}^{\min\{\text{next}(k+1,m_k)-1, K\}} a_j .$$

*Proof.* The proof follows simply by rewriting the sums several times while manipulating the definitions of $\text{prev}$ and $\text{next}$:

$$\sum_{k=1}^{K} \sum_{m \in [M] \setminus \{m_k\}} a_k b_{\text{prev}(k,m)} c_{\text{next}(k,m)}$$

$$= \sum_{j=0}^{K-1} \sum_{k=1}^{K} \sum_{m=1}^{M} a_k b_j c_{\text{next}(k,m)} \mathbb{1}_{\{m \neq m_k\}} \mathbb{1}_{\{j = \text{prev}(k,m)\}}$$

$$= b_0 \sum_{k=1}^{K} \sum_{m=1}^{M} a_k c_{\text{next}(k,m)} \mathbb{1}_{\{m \neq m_k\}} \mathbb{1}_{\{0 = \text{prev}(k,m)\}}$$

$$+ \sum_{j=1}^{K-1} \sum_{k=1}^{K} \sum_{m=1}^{M} a_k b_j c_{\text{next}(k,m)} \mathbb{1}_{\{m \neq m_k\}} \mathbb{1}_{\{j = \text{prev}(k,m)\}}$$

$$= b_0 \sum_{m=1}^{M} c_{\text{next}(1,m)} \sum_{k=1}^{\min\{\text{next}(1,m), K\}} a_k \mathbb{1}_{\{m \neq m_k\}}$$

$$+ \sum_{j=1}^{K-1} b_j c_{\text{next}(j+1,m_j)} \sum_{k=j+1}^{\min\{\text{next}(j+1,m_j), K\}} a_k \mathbb{1}_{\{m_j \neq m_k\}}$$

$$= b_0 \sum_{m=1}^{M} c_{\text{next}(1,m)} \sum_{k=1}^{\min\{\text{next}(1,m)-1, K\}} a_k + \sum_{j=1}^{K-1} b_j c_{\text{next}(j+1,m_j)} \sum_{k=j+1}^{\min\{\text{next}(j+1,m_j)-1, K\}} a_k,$$

which establishes the claim after exchanging subscripts. $\qquad\square$

## C  Proof of Theorem 2

We are ready to prove Theorem 2, which assumes that the losses are Lipschitz continuous. This assumption makes the proof shorter than that of more general Theorem 3, but on the downside, its rate has an extra $\mathcal{O}(1/K^{2/3})$ term.

**Theorem 2.** *Let $F$ be $L$-smooth, $f(\cdot; \xi)$ be $G$-Lipschitz-continuous for every $\xi$, let $\Delta \geqslant F(\mathbf{x}_0) - F^*$, and let the stochastic gradients have variance at most $\sigma^2$. Then Algorithm 1 with stepsize $\gamma = \min\left\{ \frac{1}{2ML}, \sqrt{\frac{\Delta}{L\sigma^2 K}}, \left(\frac{\Delta}{L^2 M^2 G^2 K}\right)^{1/3} \right\}$, ensures for any $K \geqslant M$*

$$\frac{1}{K} \sum_{k=1}^{K} \mathbb{E} \|\nabla F(\mathbf{x}_k)\|^2 = \mathcal{O}\left( \frac{ML\Delta}{K} + \sqrt{\frac{L\Delta\sigma^2}{K}} + \left(\frac{ML\Delta G}{K}\right)^{2/3} \right).$$

*Proof.* By the $L$-smoothness of $F$, the $\sigma^2$-bounded variance of the stochastic gradients, and the fact $\gamma \leqslant 1/(2L)$

$$\mathbb{E}\left[F(\hat{\mathbf{x}}_{k+1})\right] \leqslant \mathbb{E}\left[F(\hat{\mathbf{x}}_k) + \langle \nabla F(\hat{\mathbf{x}}_k), \hat{\mathbf{x}}_{k+1} - \hat{\mathbf{x}}_k\rangle + \frac{L}{2}\|\hat{\mathbf{x}}_{k+1} - \hat{\mathbf{x}}_k\|^2\right]$$

$$= \mathbb{E}\left[F(\hat{\mathbf{x}}_k) - \gamma\langle \nabla F(\hat{\mathbf{x}}_k), \nabla F(\mathbf{x}_k)\rangle + \frac{L\gamma^2}{2}\|\mathbf{g}_k\|^2\right]$$

$$\leqslant \mathbb{E}\left[F(\hat{\mathbf{x}}_k) + \left(\frac{L\gamma^2}{2} - \frac{\gamma}{2}\right)\|\nabla F(\mathbf{x}_k)\|^2 + \frac{\gamma}{2}\|\nabla F(\mathbf{x}_k) - \nabla F(\hat{\mathbf{x}}_k)\|^2 + \frac{L\gamma^2\sigma^2}{2}\right]$$

$$\leqslant \mathbb{E}\left[F(\hat{\mathbf{x}}_k) - \frac{\gamma}{4}\|\nabla F(\mathbf{x}_k)\|^2 + \frac{L^2\gamma}{2}\|\mathbf{x}_k - \hat{\mathbf{x}}_k\|^2 + \frac{L\gamma^2\sigma^2}{2}\right].$$

For the third term, we use Lemma 1 and the $G$-Lipschitzness of $f(\cdot; \xi)$ to bound

$$\|\mathbf{x}_k - \hat{\mathbf{x}}_k\|^2 = \left\|\sum_{m\in[M]\setminus\{m_k\}} \gamma\mathbf{g}_{\mathrm{prev}(k,m)}\right\|^2 \leqslant \gamma^2(M-1)^2 G^2.$$

Rearranging and averaging over $K$, we conclude

$$\frac{1}{K}\sum_{k=1}^K \mathbb{E}\|\nabla F(\mathbf{x}_k)\|^2 \leqslant \frac{4}{\gamma K}\sum_{k=1}^K \mathbb{E}\left[F(\hat{\mathbf{x}}_k) - F(\hat{\mathbf{x}}_{k+1}) + \frac{\gamma^3 L^2(M-1)^2 G^2}{2} + \frac{\gamma^2 L\sigma^2}{2}\right]$$

$$\leqslant \frac{4(F(\hat{\mathbf{x}}_1) - F^*)}{\gamma K} + 2\gamma^2 L^2(M-1)^2 G^2 + 2\gamma L\sigma^2.$$

All that remains is to bound $F(\hat{\mathbf{x}}_1) - F^*$. By the definition of $\hat{\mathbf{x}}_1$ from (3), the $L$-smoothness of $F$, and the fact that $\gamma \leqslant 2/(ML)$

$$\mathbb{E}[F(\hat{\mathbf{x}}_1) - F^*] \leqslant \mathbb{E}\left[F(\hat{\mathbf{x}}_0) - F^* - \gamma\sum_{m=1}^M \langle \nabla F(\mathbf{x}_0), \nabla f(\mathbf{x}_0; \xi_0^m)\rangle + \frac{\gamma^2 L}{2}\left\|\sum_{m=1}^M \nabla f(\mathbf{x}_0; \xi_0^m)\right\|^2\right]$$

$$\leqslant \mathbb{E}\left[\Delta + \left(\frac{\gamma^2 LM^2}{2} - \gamma M\right)\|\nabla F(\mathbf{x}_0)\|^2 + \frac{\gamma^2 LM\sigma^2}{2}\right] \leqslant \mathbb{E}\left[\Delta + \frac{\gamma^2 LM\sigma^2}{2}\right].$$

Substituting this into the expression above, we conclude

$$\frac{1}{K}\sum_{k=1}^K \mathbb{E}\|\nabla F(\mathbf{x}_k)\|^2 \leqslant \frac{4\Delta}{\gamma K} + 4\gamma L\sigma^2 + 2\gamma^2 L^2 M^2 G^2,$$

and plugging in the stepsize completes the proof. $\qquad\square$

We note that the expression for the stepsize comes from the idea of equalizing the terms in the last upper bound.

## D   Proof of Theorem 3

Throughout this proof, we will refer frequently to $\hat{\gamma}_1, \ldots, \hat{\gamma}_K$, the stepsizes corresponding to the stochastic gradients $\nabla f(\mathbf{x}_1, \xi_1^{m_1}), \ldots, \nabla f(\mathbf{x}_K, \xi_K^{m_K})$. However, some of these gradients will not be available when the algorithm ends after $K$ updates—in particular, precisely $M-1$ of them are in the process of being calculated when the algorithm terminates. Since those gradients are never used for updates, the corresponding stepsize $\hat{\gamma}_k$ seems, in some sense, unneeded. However, our algorithm's output weights each iterate $\mathbf{x}_k$ by a term involving $\hat{\gamma}_k$, whether or not the gradient $\nabla f(\mathbf{x}_k, \xi_k^{m_k})$ becomes available before the algorithm finishes. For this reason, in order to make the stepsize $\hat{\gamma}_k$ well-defined, we specify $\tau(k)$ for $k \geqslant K$, upon which $\hat{\gamma}_1, \ldots, \hat{\gamma}_K$ might depend, according to $\tau(k) := \max\{1, \min\{k, K\} - \mathrm{prev}(k, m_k)\}$. This is essentially equivalent to just not incrementing the iteration counter once it reaches $K$, although we note that all of the stepsizes $\hat{\gamma}_1, \ldots, \hat{\gamma}_K$ can be calculated by the time of the $K$th update, without needing to wait for the $M-1$ in-progress gradients.

## D.1 Convex case

**Lemma 2.** *In the setting of Theorem 3.1,*

$$\mathbb{E}\Big[2\sum_{k=1}^{K}\hat{\gamma}_k\,\langle\nabla f(\mathbf{x}_k;\xi_k^{m_k}),\,\mathbf{x}_k-\hat{\mathbf{x}}_k\rangle\Big]\leqslant\mathbb{E}\Big[\frac{B^2}{16}+\sum_{k=1}^{K}\hat{\gamma}_k(F(\mathbf{x}_k)-F^*)\Big].$$

*Proof.* In this proof, we denote $\mathbf{g}_k \coloneqq \nabla f(\mathbf{x}_k;\xi_k^{m_k})$ and $\zeta_k \coloneqq 2\hat{\gamma}_k\,\langle\mathbf{g}_k,\,\mathbf{x}_k-\hat{\mathbf{x}}_k\rangle$. We begin by applying Lemma 1, using the fact that for any $\mathbf{a},\mathbf{b}$, it holds $2\langle\mathbf{a},\,\mathbf{b}\rangle = \|\mathbf{a}\|^2 + \|\mathbf{b}\|^2 - \|\mathbf{a}-\mathbf{b}\|^2 \leqslant \|\mathbf{a}\|^2 + \|\mathbf{b}\|^2$, and using Assumption 1:

$$\mathbb{E}\Bigg[\sum_{k=1}^{K}\zeta_k\Bigg] = 2\mathbb{E}\Bigg[\sum_{k=1}^{K}\hat{\gamma}_k\,\langle\mathbf{g}_k,\,\mathbf{x}_k-\hat{\mathbf{x}}_k\rangle\Bigg]$$

$$= 2\mathbb{E}\Bigg[\sum_{k=1}^{K}\hat{\gamma}_k\,\Big\langle\mathbf{g}_k,\,\sum_{m\in[M]\setminus\{m_k\}}\gamma_{\text{next}(k,m)}\mathbf{g}_{\text{prev}(k,m)}\Big\rangle\Bigg]$$

$$\leqslant \mathbb{E}\Bigg[\sum_{k=1}^{K}\sum_{m\in[M]\setminus\{m_k\}}\hat{\gamma}_k^2\|\nabla F(\mathbf{x}_k)\|^2 + \gamma_{\text{next}(k,m)}^2\|\nabla F(\mathbf{x}_{\text{prev}(k,m)})\|^2\Bigg]. \qquad (11)$$

First, we bound the sum of the second terms using Lemma 4 with $a_k = 1$, $b_k = \|\nabla F(\mathbf{x}_k)\|^2$, and $c_k = \gamma_k^2$:

$$\sum_{k=1}^{K}\sum_{m\in[M]\setminus\{m_k\}}\gamma_{\text{next}(k,m)}^2\|\nabla F(\mathbf{x}_{\text{prev}(k,m)})\|^2$$

$$\leqslant \|\nabla F(\mathbf{x}_0)\|^2\sum_{m=1}^{M}\gamma_{\text{next}(1,m)}^2\min\{\text{next}(1,m)-1,K\}$$

$$+ \sum_{k=1}^{K-1}\hat{\gamma}_k^2\|\nabla F(\mathbf{x}_k)\|^2(\min\{\text{next}(k+1,m_k)-1,K\}-k). \qquad (12)$$

From here, we note that our choice of stepsize ensures

$$\gamma_k \leqslant \frac{1}{4L\tau(k)} \implies \hat{\gamma}_k \leqslant \frac{1}{4L(\min\{\text{next}(k+1,m_k),K\}-k)} \qquad (13)$$

and also $\gamma_k \leqslant \frac{1}{4LM}$ for all $k$, so substituting back into (12), we can upper bound

$$\sum_{k=1}^{K}\sum_{m\in[M]\setminus\{m_k\}}\gamma_{\text{next}(k,m)}^2\|\nabla F(\mathbf{x}_{\text{prev}(k,m)})\|^2$$

$$\leqslant \frac{\|\nabla F(\mathbf{x}_0)\|^2}{4L}\sum_{m=1}^{M}\gamma_{\text{next}(1,m)} + \frac{1}{4L}\sum_{k=1}^{K-1}\hat{\gamma}_k\|\nabla F(\mathbf{x}_k)\|^2 \qquad (14)$$

$$\leqslant \frac{B^2}{16} + \frac{1}{2}\sum_{k=1}^{K-1}\hat{\gamma}_k(F(\mathbf{x}_k)-F^*). \qquad (15)$$

Plugging this into (11), we have

$$\mathbb{E}\Bigg[\sum_{k=1}^{K}\zeta_k\Bigg] \leqslant \mathbb{E}\Bigg[\frac{B^2}{16} + \sum_{k=1}^{K}\Big(2L(M-1)\hat{\gamma}_k^2 + \frac{1}{2}\hat{\gamma}_k\Big)(F(\mathbf{x}_k)-F^*)\Bigg] \qquad (16)$$

$$\leqslant \mathbb{E}\Bigg[\frac{B^2}{16} + \sum_{k=1}^{K}\hat{\gamma}_k(F(\mathbf{x}_k)-F^*)\Bigg], \qquad (17)$$

which completes the proof. $\qquad\square$

To show fast convergence rates, we need to first show that overall the stepsizes are not too small. This is a little bit tricky because our stepsizes are delay-adaptive. Nevertheless, updates with large delays are rare, so it is natural that the majority of the stepsizes are sufficiently large.

**Lemma 5.** *In the setting of Theorem 3.1, it holds with probability one*

$$\sum_{k=1}^{K} \hat{\gamma}_k \geqslant \min\left\{ \frac{K}{36LM}, \frac{B\sqrt{K}}{3\sigma} \right\}.$$

*Proof.* First, we note that if $\tau(k) \leqslant 3M$, then

$$\gamma_k = \min\left\{ \frac{1}{12ML}, \frac{B}{\sigma\sqrt{K}} \right\}.$$

Next, we observe that if $m \neq m_k$, then $\tau(k+1, m) = \tau(k, m) + 1$, while $\tau(k+1, m_k) = 1 = 1 + \tau(k, m_k) - \tau(k)$. Therefore,

$$\sum_{m=1}^{M} \tau(k+1, m) \leqslant M - \tau(k) + \sum_{m=1}^{M} \tau(k, m).$$

Since $\sum_{m=1}^{M} \tau(1, m) = M$, we can conclude that

$$\sum_{k=1}^{K-1} \tau(k) + \sum_{m=1}^{M} \tau(K, m) \leqslant KM + \sum_{k=1}^{K-1} \tau(k) - \sum_{k=1}^{K-1} \tau(k) = KM.$$

This implies that at most $\lfloor K/3 \rfloor$ of the terms on the left hand side can be larger than $3M$. Furthermore, since the first $3M$ gradients must have delay less than $3M$, the number of terms with delay greater than $3M$ is no larger than $\min\{K/3, \max\{K - 3M, 0\}\}$. This allows us to bound

$$\sum_{k=1}^{K} \hat{\gamma}_k = \sum_{k=1}^{K-1} \gamma_k \mathbb{1}_{\{\text{prev}(k,m_k)>0\}} + \sum_{m=1}^{M} \gamma_{\text{next}(K,m)} \mathbb{1}_{\{\text{prev}(K,m)>0\}}$$

$$\geqslant \min\left\{ \frac{1}{12ML}, \frac{B}{\sigma\sqrt{K}} \right\} \left( \sum_{k=1}^{K-1} \mathbb{1}_{\{\text{prev}(k,m_k)>0\}} \mathbb{1}_{\{\tau(k,m_k)\leqslant 3M\}} \right.$$

$$\left. + \sum_{m=1}^{M} \mathbb{1}_{\{\text{prev}(K,m)>0\}} \mathbb{1}_{\{\tau(K,m)\leqslant 3M\}} \right)$$

$$\geqslant \min\left\{ \frac{1}{12LM}, \frac{B}{\sigma\sqrt{K}} \right\} \left( K - 1 - \min\left\{ \frac{K}{3}, \max\{K - 3M, 0\} \right\} \right)$$

$$\geqslant \min\left\{ \frac{1}{12LM}, \frac{B}{\sigma\sqrt{K}} \right\} \min\left\{ \frac{K}{3}, K - 1 \right\}.$$

If $K \geqslant 2$, then we conclude that

$$\sum_{k=1}^{K} \hat{\gamma}_k \geqslant \min\left\{ \frac{K}{36LM}, \frac{B\sqrt{K}}{3\sigma} \right\}$$

as claimed. Otherwise, if $K = 1$ then also $M = 1$, so all delays are one and

$$\sum_{k=1}^{K} \hat{\gamma}_k = \hat{\gamma}_1 = \min\left\{ \frac{1}{4L}, \frac{B}{\sigma} \right\} \geqslant \min\left\{ \frac{K}{36LM}, \frac{B\sqrt{K}}{3\sigma} \right\},$$

which completes the proof. $\square$

*Proof of Theorem 3.1.* We begin by expanding the update of $\hat{x}_{k+1}$ from (3):

$$\|\hat{x}_{k+1} - x^*\|^2 = \|\hat{x}_k - x^*\|^2 + \hat{\gamma}_k^2 \|g_k\|^2 - 2\hat{\gamma}_k \langle g_k, \hat{x}_k - x^* \rangle$$
$$= \|\hat{x}_k - x^*\|^2 + \hat{\gamma}_k^2 \|g_k\|^2 - 2\hat{\gamma}_k \langle g_k, x_k - x^* \rangle + 2\hat{\gamma}_k \langle g_k, x_k - \hat{x}_k \rangle.$$

From here, we take the expectation of both sides and bound the right hand side term-by-term. Because the stochastic variance is bounded by $\sigma^2$, $F$ is $L$-smooth, and Assumption 1 holds,

$$\mathbb{E}\big[\hat{\gamma}_k^2 \|\mathbf{g}_k\|^2\big] \leqslant \mathbb{E}\big[\hat{\gamma}_k^2(\|\nabla F(\mathbf{x}_k)\|^2 + \sigma^2)\big] \leqslant \mathbb{E}\big[2L\hat{\gamma}_k^2(F(\mathbf{x}_k) - F^*) + \hat{\gamma}_k^2\sigma^2)\big].$$

Likewise, by Assumption 1 and the convexity of $F$, we have

$$\mathbb{E}[\hat{\gamma}_k \langle \mathbf{g}_k, \mathbf{x}_k - \mathbf{x}^*\rangle] = \mathbb{E}[\hat{\gamma}_k \langle \nabla F(\mathbf{x}_k), \mathbf{x}_k - \mathbf{x}^*\rangle] \geqslant \mathbb{E}[\hat{\gamma}_k(F(\mathbf{x}_k) - F^*)].$$

Finally, we denote $\zeta_k := 2\hat{\gamma}_k \langle \mathbf{g}_k, \mathbf{x}_k - \hat{\mathbf{x}}_k \rangle$ and we will come back to address this term later. Combining the above inequalities, rearranging, and using that $\gamma_k \leqslant \frac{1}{4L}$ for all $k$ so $2L\hat{\gamma}_k^2 - 2\hat{\gamma}_k \leqslant \frac{3}{2}\hat{\gamma}_k$, we have

$$\frac{3}{2}\mathbb{E}[\hat{\gamma}_k(F(\mathbf{x}_k) - F^*)] \leqslant \mathbb{E}\big[\|\hat{\mathbf{x}}_k - \mathbf{x}^*\|^2 - \|\hat{\mathbf{x}}_{k+1} - \mathbf{x}^*\|^2 + \hat{\gamma}_k^2\sigma^2 + \zeta_k\big].$$

Summing over all $k$, we get

$$\frac{3}{2}\mathbb{E}\left[\sum_{k=1}^{K} \hat{\gamma}_k(F(\mathbf{x}_k) - F^*)\right] \leqslant \mathbb{E}\left[\|\hat{\mathbf{x}}_1 - \mathbf{x}^*\|^2 + \sigma^2 \sum_{k=1}^{K}\hat{\gamma}_k^2 + \sum_{k=1}^{K}\zeta_k\right].$$

So, applying Lemma 2 to the third term and rearranging, we bo

$$\mathbb{E}\left[\sum_{k=1}^{K} \hat{\gamma}_k(F(\mathbf{x}_k) - F^*)\right] \leqslant 2\mathbb{E}\left[\frac{B^2}{16} + \|\hat{\mathbf{x}}_1 - \mathbf{x}^*\|^2 + \sigma^2 \sum_{k=1}^{K}\hat{\gamma}_k^2\right]. \tag{18}$$

Applying the definition of $\hat{\mathbf{x}}_1$ from (3), and using that for convex, $L$-smooth $F$ and any $\mathbf{x}$, $\langle \nabla F(\mathbf{x}), \mathbf{x} - \mathbf{x}^*\rangle \geqslant \frac{1}{L}\|\nabla F(\mathbf{x})\|^2$, we have

$$\mathbb{E}\|\hat{\mathbf{x}}_1 - \mathbf{x}^*\|^2 = \mathbb{E}\left\|\mathbf{x}_0 - \sum_{m=1}^{M}\gamma_{\text{next}(1,m)}\nabla f(\mathbf{x}_0; \xi_0^m) - \mathbf{x}^*\right\|^2$$

$$\leqslant \mathbb{E}\left\|\mathbf{x}_0 - \sum_{m=1}^{M}\gamma_{\text{next}(1,m)}\nabla F(\mathbf{x}_0) - \mathbf{x}^*\right\|^2 + \sigma^2 \mathbb{E}\sum_{m=1}^{M}\gamma_{\text{next}(1,m)}^2$$

$$\leqslant B^2 + \mathbb{E}\left[\left(\sum_{m=1}^{M}\gamma_{\text{next}(1,m)}\right)^2 \|\nabla F(\mathbf{x}_0)\|^2\right]$$

$$- 2\mathbb{E}\left[\sum_{m=1}^{M}\gamma_{\text{next}(1,m)}\langle \nabla F(\mathbf{x}_0), \mathbf{x}_0 - \mathbf{x}^*\rangle\right] + \sigma^2\mathbb{E}\sum_{m=1}^{M}\gamma_{\text{next}(1,m)}^2$$

$$\leqslant B^2 + \mathbb{E}\left[\left(-\frac{2}{L} + \sum_{m=1}^{M}\gamma_{\text{next}(1,m)}\right)\left(\sum_{m=1}^{M}\gamma_{\text{next}(1,m)}\right)\|\nabla F(\mathbf{x}_0)\|^2\right]$$

$$+ \sigma^2\mathbb{E}\sum_{m=1}^{M}\gamma_{\text{next}(1,m)}^2$$

$$\leqslant B^2 + \sigma^2\mathbb{E}\sum_{m=1}^{M}\gamma_{\text{next}(1,m)}^2, \tag{19}$$

because $\sum_{m=1}^{M}\gamma_{\text{next}(1,m)} \leqslant \frac{1}{4L}$. Returning to (18), for

$$\tilde{\mathbf{x}}_K := \frac{1}{\sum_{k=1}^{K}\hat{\gamma}_k}\sum_{k=1}^{K}\hat{\gamma}_k\mathbf{x}_k \tag{20}$$

we conclude by the convexity of $F$ that

$$\mathbb{E}[F(\tilde{\mathbf{x}}_K) - F^*] \leqslant \mathbb{E}\left[\frac{1}{\sum_{k=1}^{K}\hat{\gamma}_k}\sum_{k=1}^{K}\hat{\gamma}_k(F(\mathbf{x}_k) - F^*)\right]$$

$$\leqslant \mathbb{E}\left[\frac{3B^2}{\sum_{k=1}^{K}\hat{\gamma}_k} + \frac{2\sigma^2\sum_{k=1}^{K}\hat{\gamma}_k^2}{\sum_{k=1}^{K}\hat{\gamma}_k}\right].$$

From here, all that remains is to observe that

$$2\sigma^2 \sum_{k=1}^{K} \hat{\gamma}_k^2 \leqslant 2\sigma^2 \sum_{k=1}^{K} \left(\frac{B}{\sigma\sqrt{K}}\right)^2 = 2B^2$$

and by Lemma 5

$$\sum_{k=1}^{K} \hat{\gamma}_k \geqslant \min\left\{\frac{K}{36ML}, \frac{B\sqrt{K}}{3\sigma}\right\}.$$

So, we can conclude that

$$\mathbb{E}[F(\tilde{\mathbf{x}}_K) - F^*] \leqslant \mathbb{E}\left[\frac{5B^2}{\sum_{k=1}^{K} \hat{\gamma}_k}\right] \leqslant \frac{5B^2}{\min\left\{\frac{K}{36ML}, \frac{B\sqrt{K}}{3\sigma}\right\}} \leqslant \frac{180MLB^2}{K} + \frac{15\sigma B}{\sqrt{K}},$$

which completes the proof. $\qquad\square$

### D.2   Strongly convex case

**Lemma 6.** *In the setting of Theorem 3.2, with $\gamma_{\max}$ defined as in (21),*

$$\mathbb{E}\left[2L\sum_{k=1}^{K} \hat{\gamma}_k \hat{P}_k \|\mathbf{x}_k - \hat{\mathbf{x}}_k\|^2\right] \leqslant \frac{\sigma^2 M \gamma_{\max}^2}{2} + \frac{B^2}{32} + \mathbb{E}\left[\frac{\sigma^2 \gamma_{\max}}{2} \sum_{k=1}^{K} \hat{\gamma}_k \hat{P}_k + \frac{1}{4}\sum_{k=1}^{K} \hat{\gamma}_k \hat{P}_k F_k\right].$$

*Proof.* We start by applying Lemma 3 and then Lemma 4 with $a_k = \hat{\gamma}_k \hat{P}_k$, $b_k = \sigma^2 + (M-1)\|\nabla F(\mathbf{x}_k)\|^2$, and $c_k = \gamma_k^2$:

$$\mathbb{E}\left[2L\sum_{k=1}^{K} \hat{\gamma}_k \hat{P}_k \|\mathbf{x}_k - \hat{\mathbf{x}}_k\|^2\right]$$

$$\leqslant 4L\mathbb{E}\left[\sum_{k=1}^{K}\sum_{m\neq m_k} a_k \gamma_{\text{next}(k,m)}^2 \left[\sigma^2 + (M-1)\|\nabla F(\mathbf{x}_{\text{prev}(k,m)})\|^2\right]\right]$$

$$\leqslant 4L\mathbb{E}\left[\left(\sigma^2 + (M-1)\|\nabla F(\mathbf{x}_0)\|^2\right) \sum_{m=1}^{M} \gamma_{\text{next}(1,m)}^2 \sum_{j=1}^{\min\{\text{next}(1,m)-1,\,K\}} a_j\right.$$

$$\left.+ \sum_{k=1}^{K-1} \left(\sigma^2 + (M-1)\|\nabla F(\mathbf{x}_k)\|^2\right)\gamma_{\text{next}(k+1,m_k)}^2 \sum_{j=k+1}^{\min\{\text{next}(k+1,m_k)-1,\,K\}} a_j\right]$$

$$\leqslant 4L\mathbb{E}\left[\left(\sigma^2 + 2LMF_0\right) \sum_{m=1}^{M} \gamma_{\text{next}(1,m)}^2 \sum_{j=1}^{\min\{\text{next}(1,m)-1,\,K\}} a_j\right.$$

$$\left.+ \sum_{k=1}^{K-1} \left(\sigma^2 + 2LMF_k\right)\hat{\gamma}_k^2 \sum_{j=k+1}^{\min\{\text{next}(k+1,m_k)-1,\,K\}} a_j\right].$$

Since the sequence $\hat{P}_k$ is increasing, for each $k$ we can bound $a_k = \hat{\gamma}_k \hat{P}_k \leqslant \gamma_{\max} \hat{P}_j$ for any $j \geqslant k$. So, denoting $n_{1,m} := \min\{\text{next}(1,m)-1, K\}$ and $n_k = \min\{\text{next}(k+1,m_k)-1, K\}$ for short, we have

$$\mathbb{E}\left[2L\sum_{k=1}^{K} \hat{\gamma}_k \hat{P}_k \|\mathbf{x}_k - \hat{\mathbf{x}}_k\|^2\right] \leqslant 4L\mathbb{E}\left[\left(\sigma^2 + 2LMF_0\right) \sum_{m=1}^{M} \gamma_{\text{next}(1,m)}^2 n_{1,m}\gamma_{\max}\hat{P}_{n_{1,m}}\right.$$

$$\left.+ \sum_{k=1}^{K-1} \left(\sigma^2 + 2LMF_k\right)\hat{\gamma}_k^2 (n_k - k)\gamma_{\max}\hat{P}_{n_k}\right].$$

We also have that for $j \geqslant k$

$$\frac{\hat{P}_j}{\hat{P}_k} = \exp\left(\mu \sum_{i=k+1}^{j} \hat{\gamma}_i\right) \leqslant e^{\mu\gamma_{\max}(j-k)},$$

so,

$$\mathbb{E}\left[2L\sum_{k=1}^{K} \hat{\gamma}_k \hat{P}_k \|\mathbf{x}_k - \hat{\mathbf{x}}_k\|^2\right] \leqslant 4L\gamma_{\max}\mathbb{E}\Bigg[\left(\sigma^2 + 2LMF_0\right)\sum_{m=1}^{M} \gamma_{\text{next}(1,m)}^2 n_{1,m} e^{\mu\gamma_{\max}n_{1,m}}$$
$$+ \sum_{k=1}^{K-1} \left(\sigma^2 + 2LMF_k\right)\hat{\gamma}_k^2 (n_k - k)\hat{P}_k e^{\mu\gamma_{\max}(n_k-k)}\Bigg].$$

We now recall our choice of stepsize

$$\gamma_k \leqslant \frac{\exp\left(-\frac{\mu\tau(k)}{4ML}\right)}{4L\tau(k)} \leqslant \frac{\exp(-\mu\gamma_{\max}\tau(k))}{4L\tau(k)},$$

which implies for each $j$ that

$$\hat{\gamma}_j \leqslant \frac{1}{4L(n_j - j + 1)}\exp(-\mu\gamma_{\max}(n_j - j + 1))$$

and

$$\gamma_{\text{next}(1,m)} \leqslant \frac{1}{4Ln_{1,m}}\exp(-\mu\gamma_{\max}(n_{1,m} + 1)).$$

Therefore,

$$\mathbb{E}\left[2L\sum_{k=1}^{K} \hat{\gamma}_k \hat{P}_k \|\mathbf{x}_k - \hat{\mathbf{x}}_k\|^2\right]$$
$$\leqslant \frac{\gamma_{\max}}{2}\mathbb{E}\left[\left(\sigma^2 + L^2MB^2\right)\sum_{m=1}^{M} \gamma_{\text{next}(1,m)} + \sum_{k=1}^{K-1}\left(\sigma^2 + 2LMF_k\right)\hat{\gamma}_k \hat{P}_k\right]$$
$$\leqslant \frac{\sigma^2 M\gamma_{\max}^2}{2} + \frac{B^2}{32} + \mathbb{E}\left[\frac{\sigma^2\gamma_{\max}}{2}\sum_{k=1}^{K-1} \hat{\gamma}_k \hat{P}_k + \frac{1}{4}\sum_{k=1}^{K-1} \hat{\gamma}_k \hat{P}_k F_k\right],$$

where we used that $\gamma_{\max} \leqslant 1/(8ML)$ and $F_0 \leqslant \frac{1}{2}LB^2$. $\qquad\square$

**Lemma 7.** *In the setting of Theorem 3.2, with $\gamma_{\max}$ defined as in (21), we have with probability 1*

$$\sum_{k=1}^{K} \hat{\gamma}_k \hat{P}_k \geqslant \max\left\{\frac{K\gamma_{\max}}{42}, \ \frac{\gamma_{\max}}{7}\exp\left(\frac{K\mu\gamma_{\max}}{504}\right)\right\}.$$

*Proof.* First, we note that if $\tau(k) \leqslant 3M$, then since $\gamma_{\max} \leqslant \frac{1}{4LM}$:

$$\gamma_k \geqslant \min\left\{\gamma_{\max} \frac{\exp\left(-\frac{3\mu}{4L}\right)}{12ML}\right\} \geqslant \frac{\gamma_{\max}}{7}.$$

As in the proof of Lemma 5, we recall that $\tau(k,m)$ denotes the current delay of the $m$th worker's gradient. We begin by observing that if $m \neq m_k$, then $\tau(k+1,m) = \tau(k,m)+1$ and $\tau(k+1,m_k) = 1 \leqslant \tau(k,m_k)$. Therefore,

$$\sum_{m=1}^{M} \tau(k+1,m) \leqslant M - 1 + \sum_{m=1}^{M} \tau(k,m).$$

Since $\sum_{m=1}^{M} \tau(1,m) = M$, we can conclude that for every $J \leqslant K$

$$\sum_{k=1}^{J} \tau(k) \leqslant \sum_{k=1}^{J}\sum_{m=1}^{M} \tau(k,m) \leqslant JM.$$

This implies that at most $\lfloor J/3 \rfloor$ of the terms on the left hand side can be larger than $3M$. Furthermore, since the first $3M$ gradients must have delay less than $3M$, the number of terms with delay greater than $3M$ is no larger than $\min\{J/3, \max\{J - 3M, 0\}\}$. This allows us to bound for each $J \leqslant K$

$$
\begin{aligned}
\sum_{k=1}^{J} \hat{\gamma}_k &\geqslant \sum_{k=1}^{J} \gamma_k \mathbb{1}_{\{\text{prev}(k,m_k)>0\}} \\
&\geqslant \sum_{k=1}^{J} \gamma_k \mathbb{1}_{\{\text{prev}(k,m_k)>0\}} \mathbb{1}_{\{\tau(k)\leqslant 3M\}} \\
&\geqslant \frac{\gamma_{\max}}{7} \sum_{k=1}^{J} \mathbb{1}_{\{\text{prev}(k,m_k)>0\}} \mathbb{1}_{\{\tau(k)\leqslant 3M\}} \\
&\geqslant \frac{\gamma_{\max}}{7} \left( J - M - \min\left\{ \frac{J}{3}, \max\{J - 3M, 0\} \right\} \right) \\
&= \frac{\gamma_{\max}}{7} \max\left\{ \frac{2J}{3} - M, \min\{2M, J - M\} \right\} \\
&\geqslant \max\left\{ \frac{\gamma_{\max}(J - M)}{14}, 0 \right\}.
\end{aligned}
$$

With this in hand, we again observe

$$
\sum_{k=1}^{K-1} \tau(k) + \sum_{m=1}^{M} \tau(K,m) \leqslant \sum_{k=1}^{K}\sum_{m=1}^{M} \tau(k,m) \leqslant KM.
$$

This implies that at most $\lfloor K/3 \rfloor$ of the terms on the left hand side can be larger than $3M$. Furthermore, since the first $3M$ gradients must have delay less than $3M$, the number of terms with delay greater than $3M$ is no larger than $\min\{K/3, \max\{K - 3M, 0\}\}$. So,

$$
\begin{aligned}
\sum_{k=1}^{K} \hat{\gamma}_k \hat{P}_k &\geqslant \sum_{k=1}^{K-1} \gamma_k \hat{P}_{\text{prev}(k,m_k)} \mathbb{1}_{\{\text{prev}(k,m_k)>0\}} + \sum_{m=1}^{M} \gamma_{\text{next}(K,m)} \hat{P}_{\text{prev}(K,m)} \mathbb{1}_{\{\text{prev}(K,m)>0\}} \\
&\geqslant \frac{\gamma_{\max}}{7} \Bigg( \sum_{k=1}^{K-1} \hat{P}_{\text{prev}(k,m_k)} \mathbb{1}_{\{\text{prev}(k,m_k)>0\}} \mathbb{1}_{\{\tau(k)\leqslant 3M\}} \\
&\qquad\qquad\qquad\qquad + \sum_{m=1}^{M} \hat{P}_{\text{prev}(K,m)} \mathbb{1}_{\{\text{prev}(K,m)>0\}} \mathbb{1}_{\{\tau(K,m)\leqslant 3M\}} \Bigg) \\
&\geqslant \frac{\gamma_{\max}}{7} \min_{\iota_1,\dots,\iota_K \in \{0,1\}} \left\{ \sum_{k=1}^{K} \hat{P}_k(1 - \iota_k) \quad \text{s.t.} \quad \sum_{k=1}^{K} \iota_k \leqslant \min\left\{ \frac{K}{3}, \max\{K - 3M, 0\} \right\} \right\} \\
&\geqslant \frac{\gamma_{\max}}{7} \sum_{k=1}^{2K/3} \hat{P}_k = \frac{\gamma_{\max}}{7} \sum_{k=1}^{2K/3} \exp\left( \mu \sum_{j=1}^{k} \hat{\gamma}_j \right) \\
&\geqslant \frac{\gamma_{\max}}{7} \sum_{k=1}^{2K/3} \exp\left( \frac{(k - M)\mu\gamma_{\max}}{14} \right) \\
&\geqslant \frac{\gamma_{\max}}{7} \sum_{k=1}^{K/6} \exp\left( \frac{k\mu\gamma_{\max}}{84} \right) \\
&\geqslant \max\left\{ \frac{K\gamma_{\max}}{42}, \frac{\gamma_{\max}}{7} \exp\left( \frac{K\mu\gamma_{\max}}{504} \right) \right\},
\end{aligned}
$$

where we used that $\hat{P}_k$ is increasing, then the lower bound on the sum of stepsizes from above, and then that $K \geqslant 3M$ so $k - M \geqslant k/6$ for $k \geqslant K/2$. $\qquad\square$

*Proof of Theorem 3.2.* Throughout the proof, we will use $\gamma_{\max}$ to denote the $\tau(k)$-independent portion of the stepsize, i.e.,

$$\gamma_{\max} := \min\left\{\frac{1}{8ML}, \frac{504\log\left(e + \frac{\mu^2 K^2 B^2}{\sigma^2}\right)}{\mu K}\right\}. \tag{21}$$

Mirroring the typical analysis of SGD, we begin by recalling the updates of $\hat{\mathbf{x}}_{k+1}$ from (3) and expanding:

$$
\begin{aligned}
\mathbb{E}\|\hat{\mathbf{x}}_{k+1} - \mathbf{x}^*\|^2 &= \mathbb{E}\|\hat{\mathbf{x}}_k - \hat{\gamma}_k \nabla f(\mathbf{x}_k; \xi_k^{m_k}) - \mathbf{x}^*\|^2 \\
&= \mathbb{E}\left[\|\hat{\mathbf{x}}_k - \mathbf{x}^*\|^2 + \hat{\gamma}_k^2 \|\nabla f(\mathbf{x}_k; \xi_k^{m_k})\|^2 - 2\hat{\gamma}_k \left\langle \nabla f(\mathbf{x}_k; \xi_k^{m_k}), \hat{\mathbf{x}}_k - \mathbf{x}^*\right\rangle\right] \\
&\leqslant \mathbb{E}\left[\|\hat{\mathbf{x}}_k - \mathbf{x}^*\|^2 + \hat{\gamma}_k^2 \sigma^2 + \hat{\gamma}_k^2 \|\nabla F(\mathbf{x}_k)\|^2 - 2\hat{\gamma}_k \left\langle \nabla F(\mathbf{x}_k), \mathbf{x}_k - \mathbf{x}^* + \hat{\mathbf{x}}_k - \mathbf{x}_k\right\rangle\right].
\end{aligned}
$$

For the inequality, we used Assumption 1 and the $\sigma^2$ stochastic gradient variance bound. To proceed, the L-smoothness of $F$ implies $\|\nabla F(\mathbf{x}_k)\|^2 \leqslant 2L(F(\mathbf{x}_k) - F^*)$, and to bound the final term, we use the $\mu$-strong convexity of $F$ and the inequality $2\langle \mathbf{a}, \mathbf{b}\rangle \leqslant \alpha\|\mathbf{a}\|^2 + \alpha^{-1}\|\mathbf{b}\|^2$ for any $\alpha > 0$, so:

$$
\begin{aligned}
-2\hat{\gamma}_k &\langle \nabla F(\mathbf{x}_k), \mathbf{x}_k - \mathbf{x}^* + \hat{\mathbf{x}}_k - \mathbf{x}_k\rangle \\
&\leqslant -2\hat{\gamma}_k\left(F(\mathbf{x}_k) - F^* + \frac{\mu}{2}\|\mathbf{x}_k - \mathbf{x}^*\|^2\right) + \hat{\gamma}_k\left(\frac{1}{2L}\|\nabla F(\mathbf{x}_k)\|^2 + 2L\|\mathbf{x}_k - \hat{\mathbf{x}}_k\|^2\right) \\
&\leqslant -\hat{\gamma}_k(F(\mathbf{x}_k) - F^*) - \mu\hat{\gamma}_k\|\mathbf{x}_k - \mathbf{x}^*\|^2 + 2L\hat{\gamma}_k\|\mathbf{x}_k - \hat{\mathbf{x}}_k\|^2.
\end{aligned}
$$

Combining this with the above, and using that $\gamma_k \leqslant \frac{1}{4L}$ for all $k$, so $2L\hat{\gamma}_k^2 - \hat{\gamma}_k \leqslant \frac{1}{2}\hat{\gamma}_k$, we conclude

$$\mathbb{E}\|\hat{\mathbf{x}}_{k+1} - \mathbf{x}^*\|^2 \leqslant \mathbb{E}\left[(1 - \mu\hat{\gamma}_k)\|\hat{\mathbf{x}}_k - \mathbf{x}^*\|^2 + \hat{\gamma}_k^2\sigma^2 - \frac{\hat{\gamma}_k}{2}F_k + 2L\hat{\gamma}_k\|\mathbf{x}_k - \hat{\mathbf{x}}_k\|^2\right].$$

where $F_k := F(\mathbf{x}_k) - F^*$. From here, we define a weighting of the iterates

$$\hat{P}_k := \exp\left(-\mu \sum_{j=1}^{k} \hat{\gamma}_j\right),$$

we multiply both sides of the expression by $\hat{P}_k$, sum over $k$, and use that $1 - \mu\hat{\gamma}_k \leqslant \exp(-\mu\hat{\gamma}_k)$ in order to telescope the sum, getting:

$$
\begin{aligned}
\mathbb{E}&\left[\frac{1}{2}\sum_{k=1}^{K} \hat{\gamma}_k \hat{P}_k F_k\right] \\
&\leqslant \mathbb{E}\left[\sum_{k=1}^{K} \hat{P}_k\left((1 - \mu\hat{\gamma}_k)\|\hat{\mathbf{x}}_k - \mathbf{x}^*\|^2 - \|\hat{\mathbf{x}}_{k+1} - \mathbf{x}^*\|^2 + \hat{\gamma}_k^2\sigma^2 + 2L\hat{\gamma}_k\|\mathbf{x}_k - \hat{\mathbf{x}}_k\|^2\right)\right] \\
&\leqslant \mathbb{E}\left[\|\hat{\mathbf{x}}_1 - \mathbf{x}^*\|^2 + \sum_{k=2}^{K}\left[(1 - \mu\hat{\gamma}_k)\hat{P}_k - \hat{P}_{k-1}\right]\|\hat{\mathbf{x}}_k - \mathbf{x}^*\|^2 + \sum_{k=1}^{K}\hat{P}_k\left(\hat{\gamma}_k^2\sigma^2 + 2L\hat{\gamma}_k\|\mathbf{x}_k - \hat{\mathbf{x}}_k\|^2\right)\right] \\
&\leqslant \mathbb{E}\left[\|\hat{\mathbf{x}}_1 - \mathbf{x}^*\|^2 + \sigma^2\sum_{k=1}^{K}\hat{\gamma}_k^2\hat{P}_k + 2L\sum_{k=1}^{K}\hat{\gamma}_k\hat{P}_k\|\mathbf{x}_k - \hat{\mathbf{x}}_k\|^2\right].
\end{aligned}
$$

We apply Lemma 6 and rearrange, giving

$$\sum_{k=1}^{K} \hat{\gamma}_k \hat{P}_k F_k \leqslant \mathbb{E}\left[\frac{B^2}{8} + 4\|\hat{\mathbf{x}}_1 - \mathbf{x}^*\|^2 + 2\sigma^2 M\gamma_{\max}^2 + 6\sigma^2\gamma_{\max}\sum_{k=1}^{K}\hat{\gamma}_k\hat{P}_k\right].$$

In the proof of the smooth convex case of Theorem 3, we derived the following upper bound in (19):

$$\mathbb{E}\|\hat{\mathbf{x}}_1 - \mathbf{x}^*\|^2 \leqslant \mathbb{E}\left[\|\mathbf{x}_0 - \mathbf{x}^*\|^2 + \sigma^2\sum_{m=1}^{M}\gamma_{\text{next}(1,m)}^2\right] \leqslant B^2 + \sigma^2 M\gamma_{\max}^2.$$

Therefore, for
$$\tilde{\mathbf{x}}_K := \frac{\sum_{k=1}^K \hat{\gamma}_k \hat{P}_k \mathbf{x}_k}{\sum_{k=1}^K \hat{\gamma}_k \hat{P}_k}, \tag{22}$$
we conclude by the convexity of $F$ that
$$\mathbb{E}[F(\tilde{\mathbf{x}}_K) - F^*] \leqslant \mathbb{E}\left[\frac{\sum_{k=1}^K \hat{\gamma}_k \hat{P}_k F_k}{\sum_{k=1}^K \hat{\gamma}_k \hat{P}_k}\right] \leqslant \mathbb{E}\left[\frac{5B^2 + 6\sigma^2 M \gamma_{\max}^2}{\sum_{k=1}^K \hat{\gamma}_k \hat{P}_k} + 6\sigma^2 \gamma_{\max}\right].$$
To conclude, we use Lemma 7 to lower bound the denominator of the first term:
$$\mathbb{E}[F(\tilde{\mathbf{x}}_K) - F^*] \leqslant \frac{5B^2 + 6\sigma^2 M \gamma_{\max}^2}{\max\left\{\frac{K\gamma_{\max}}{42}, \; \frac{\gamma_{\max}}{7} \exp\left(\frac{K\mu\gamma_{\max}}{504}\right)\right\}} + 6\sigma^2 \gamma_{\max}$$
$$\leqslant \frac{35B^2}{\gamma_{\max}} \exp\left(-\frac{K\mu\gamma_{\max}}{504}\right) + 252\sigma^2 \gamma_{\max}.$$
Plugging in our choice of $\gamma_{\max}$ from (21) completes the proof. $\qquad\square$

### D.3 Non-convex case

**Lemma 8.** *In the setting of Theorem 3.3, with $\gamma_{\max}$ defined as in (24)*
$$\mathbb{E}\left[\frac{L^2}{2} \sum_{k=1}^K \hat{\gamma}_k \|\mathbf{x}_k - \hat{\mathbf{x}}_k\|^2\right] \leqslant \mathbb{E}\left[\frac{\Delta}{8} + \frac{L\sigma^2 \gamma_{\max}}{4}\left(M\gamma_{\max} + \sum_{k=1}^K \hat{\gamma}_k\right) + \frac{1}{8}\sum_{k=1}^K \hat{\gamma}_k \|\nabla F(\mathbf{x}_k)\|^2\right].$$

*Proof.* We start using Lemma 3 and then Lemma 4 with $a_k = \hat{\gamma}_k$, $b_k = \sigma^2 + (M-1)\|\nabla F(\mathbf{x}_k)\|^2$, and $c_k = \gamma_k^2$:
$$\mathbb{E}\left[\frac{L^2}{2} \sum_{k=1}^K \hat{\gamma}_k \|\mathbf{x}_k - \hat{\mathbf{x}}_k\|^2\right]$$
$$\leqslant L^2 \mathbb{E}\left[\sum_{k=1}^K \hat{\gamma}_k \sum_{m\in[M]\setminus\{m_k\}} \gamma_{\text{next}(k,m)}^2 \left[\sigma^2 + (M-1)\|\nabla F(\mathbf{x}_{\text{prev}(k,m)})\|^2\right]\right]$$
$$\leqslant L^2 \gamma_{\max} \mathbb{E}\left[\left(\sigma^2 + (M-1)\|\nabla F(\mathbf{x}_0)\|^2\right) \sum_{m=1}^M \gamma_{\text{next}(1,m)}^2 \min\{\text{next}(1,m) - 1, K\}\right.$$
$$\left. + \sum_{k=1}^{K-1} \left(\sigma^2 + (M-1)\|\nabla F(\mathbf{x}_k)\|^2\right) \hat{\gamma}_k^2 (\min\{\text{next}(k+1, m_k) - 1, K\} - k)\right].$$
From here, note that our choice of stepsize
$$\gamma_k \leqslant \frac{1}{4L\tau(k)}$$
implies
$$\gamma_{\text{next}(1,m)} \min\{\text{next}(1,m) - 1, K\} \leqslant \frac{1}{4L},$$
$$\hat{\gamma}_k (\min\{\text{next}(k+1, m_k) - 1, K\} - k) \leqslant \frac{1}{4L}.$$
Plugging these two inequalities into our previous bound, we obtain
$$\mathbb{E}\left[\frac{L^2}{2} \sum_{k=1}^K \hat{\gamma}_k \|\mathbf{x}_k - \hat{\mathbf{x}}_k\|^2\right]$$
$$\leqslant \frac{L\gamma_{\max}}{4} \mathbb{E}\left[\left(\sigma^2 + M\|\nabla F(\mathbf{x}_0)\|^2\right) \sum_{m=1}^M \gamma_{\text{next}(1,m)} + \sum_{k=1}^{K-1} \left(\sigma^2 + M\|\nabla F(\mathbf{x}_k)\|^2\right) \hat{\gamma}_k\right]$$
$$\leqslant \mathbb{E}\left[\frac{ML\gamma_{\max}^2}{4}\left(\sigma^2 + M\|\nabla F(\mathbf{x}_0)\|^2\right) + \frac{L\sigma^2 \gamma_{\max}}{4} \sum_{k=1}^K \hat{\gamma}_k + \frac{ML\gamma_{\max}}{4} \sum_{k=1}^K \hat{\gamma}_k \|\nabla F(\mathbf{x}_k)\|^2\right].$$

Using the fact that $\gamma_{\max} \leqslant 1/(2ML)$ and $\|\nabla F(x_0)\|^2 \leqslant 2L\Delta$ completes the proof. $\qquad\square$

**Lemma 9.** *In the setting of Theorem 3.3, with $\gamma_{\max}$ defined as in* (24)

$$\sum_{k=1}^{K} \hat{\gamma}_k \geqslant \frac{K\gamma_{\max}}{9}.$$

*Proof.* First, we note that if $\tau(k) \leqslant 3M$, then

$$\gamma_k = \min\left\{ \frac{1}{6L}, \frac{1}{2ML}, \sqrt{\frac{\Delta}{KL\sigma^2}} \right\} \geqslant \frac{\gamma_{\max}}{3}.$$

Following the proofs of Lemmas 5 and 7, we begin with the observation that

$$\sum_{k=1}^{K-1} \tau(k, m_k) + \sum_{m=1}^{M} \tau(K, m) \leqslant \sum_{k=1}^{K}\sum_{m=1}^{M} \tau(k, m) \leqslant KM. \tag{23}$$

This implies that at most $\lfloor K/3 \rfloor$ of the terms on the left hand side can be larger than $3M$. Furthermore, since the first $3M$ gradients must have delay less than $3M$, the number of terms with delay greater than $3M$ is no larger than $\min\{K/3, \max\{K - 3M, 0\}\}$.

From here, we rewrite:

$$\sum_{k=1}^{K} \hat{\gamma}_k = \sum_{k=1}^{K-1} \gamma_k \mathbb{1}_{\{\text{prev}(k,m_k)>0\}} + \sum_{m=1}^{M} \gamma_{\text{next}(K,m)} \mathbb{1}_{\{\text{prev}(K,m)>0\}}$$

$$\geqslant \sum_{k=1}^{K-1} \gamma_k \mathbb{1}_{\{\text{prev}(k,m_k)>0\}} \mathbb{1}_{\{\tau(k,m_k)\leqslant 3M\}} + \sum_{m=1}^{M} \gamma_{\text{next}(K,m)} \mathbb{1}_{\{\text{prev}(K,m)>0\}} \mathbb{1}_{\{\tau(K,m)\leqslant 3M\}}$$

$$\geqslant \frac{\gamma_{\max}}{3}\left( \sum_{k=1}^{K-1} \mathbb{1}_{\{\text{prev}(k,m_k)>0\}} \mathbb{1}_{\{\tau(k,m_k)\leqslant 3M\}} + \sum_{m=1}^{M} \mathbb{1}_{\{\text{prev}(K,m)>0\}} \mathbb{1}_{\{\tau(K,m)\leqslant 3M\}} \right)$$

$$\geqslant \frac{\gamma_{\max}}{3}\left( K + M - 1 - \left( \min\left\{ \frac{K}{3}, \max\{K - 3M, 0\} \right\} + M \right) \right)$$

$$= \frac{\gamma_{\max}}{3} \max\left\{ \frac{2K}{3} - 1, \min\{3M - 1, K - 1\} \right\}$$

$$\geqslant \frac{\gamma_{\max}}{3} \min\left\{ \frac{K}{3}, K - 1 \right\}.$$

For $K \geqslant 2$, the Lemma follows directly. For $K = 1$, we also have $M = 1$ so all of the delays are one, and $\sum_{k=1}^{K} \hat{\gamma}_k = \hat{\gamma}_1 = \min\{\gamma_{\max}, \frac{1}{2L}\} \geqslant \frac{K\gamma_{\max}}{9}$. Therefore, this inequality holds either way, completing the proof. $\qquad\square$

*Proof of Theorem 3.3.* In this proof, we will use $\gamma_{\max}$ to denote the $\tau(k)$-independent terms in the definition of the stepsize, i.e.,

$$\gamma_{\max} := \min\left\{ \frac{1}{2ML}, \sqrt{\frac{\Delta}{KL\sigma^2}} \right\}. \tag{24}$$

Next, we use the $L$-smoothness of $F$, the definition of $\hat{\mathbf{x}}_{k+1}$ from (3), and Assumption 1:

$$\mathbb{E}F(\hat{\mathbf{x}}_{k+1}) \leqslant \mathbb{E}\left[ F(\hat{\mathbf{x}}_k) + \langle \nabla F(\hat{\mathbf{x}}_k), \hat{\mathbf{x}}_{k+1} - \hat{\mathbf{x}}_k \rangle + \frac{L}{2}\|\hat{\mathbf{x}}_{k+1} - \hat{\mathbf{x}}_k\|^2 \right]$$

$$= \mathbb{E}\left[ F(\hat{\mathbf{x}}_k) - \hat{\gamma}_k \langle \nabla F(\hat{\mathbf{x}}_k), \nabla F(\mathbf{x}_k) \rangle + \frac{L\hat{\gamma}_k^2}{2}\|\nabla f(\mathbf{x}_k; \xi_k^{m_k})\|^2 \right]$$

$$\leqslant \mathbb{E}\left[ F(\hat{\mathbf{x}}_k) + \left( \frac{L\hat{\gamma}_k^2}{2} - \frac{\hat{\gamma}_k}{2} \right)\|\nabla F(\mathbf{x}_k)\|^2 + \frac{\hat{\gamma}_k}{2}\|\nabla F(\hat{\mathbf{x}}_k) - \nabla F(\mathbf{x}_k)\|^2 + \frac{L\sigma^2 \hat{\gamma}_k^2}{2} \right]$$

$$\leqslant \mathbb{E}\left[ F(\hat{\mathbf{x}}_k) + \left( \frac{L\hat{\gamma}_k^2}{2} - \frac{\hat{\gamma}_k}{2} \right)\|\nabla F(\mathbf{x}_k)\|^2 + \frac{L^2\hat{\gamma}_k}{2}\|\mathbf{x}_k - \hat{\mathbf{x}}_k\|^2 + \frac{L\sigma^2 \hat{\gamma}_k^2}{2} \right].$$

Since $\gamma_k \leqslant \frac{1}{2L}$ for all $k$, this means

$$\mathbb{E}[F(\hat{\mathbf{x}}_{k+1}) - F(\hat{\mathbf{x}}_k)] \leqslant \mathbb{E}\left[-\frac{\hat{\gamma}_k}{4}\|\nabla F(\mathbf{x}_k)\|^2 + \frac{L^2\hat{\gamma}_k}{2}\|\mathbf{x}_k - \hat{\mathbf{x}}_k\|^2 + \frac{L\sigma^2\hat{\gamma}_k^2}{2}\right].$$

Rearranging and summing over $k$, this means

$$\frac{1}{4}\mathbb{E}\left[\sum_{k=1}^K \hat{\gamma}_k\|\nabla F(\mathbf{x}_k)\|^2\right] \leqslant \mathbb{E}\left[F(\hat{\mathbf{x}}_1) - F(\hat{\mathbf{x}}_{K+1}) + \frac{L^2}{2}\sum_{k=1}^K \hat{\gamma}_k\|\mathbf{x}_k - \hat{\mathbf{x}}_k\|^2 + \frac{L\sigma^2}{2}\sum_{k=1}^K \hat{\gamma}_k^2\right]$$

$$\leqslant \mathbb{E}\left[\Delta + \frac{L^2}{2}\sum_{k=1}^K \hat{\gamma}_k\|\mathbf{x}_k - \hat{\mathbf{x}}_k\|^2 + \frac{L\sigma^2}{2}\sum_{k=1}^K \hat{\gamma}_k^2\right].$$

Applying Lemma 8 and rearranging, this gives

$$\mathbb{E}\left[\sum_{k=1}^K \hat{\gamma}_k\|\nabla F(\mathbf{x}_k)\|^2\right] \leqslant \mathbb{E}\left[9\Delta + 2ML\sigma^2\gamma_{\max}^2 + 6L\sigma^2\gamma_{\max}\sum_{k=1}^K \hat{\gamma}_k\right].$$

Therefore, if we choose an output vector $\tilde{\mathbf{x}}_K$ with

$$\mathbb{P}(\tilde{\mathbf{x}}_K = \mathbf{x}_k) \propto \hat{\gamma}_k \quad \forall k \in [K], \tag{25}$$

then by Lemma 9,

$$\mathbb{E}\left[\|\nabla F(\tilde{\mathbf{x}}_K)\|^2\right] \leqslant \mathbb{E}\left[\frac{9\Delta + 2ML\sigma^2\gamma_{\max}^2}{\sum_{k=1}^K \hat{\gamma}_k} + 6L\sigma^2\gamma_{\max}\right]$$

$$\leqslant \mathbb{E}\left[\frac{81\Delta + 18ML\sigma^2\gamma_{\max}^2}{K\gamma_{\max}} + 6L\sigma^2\gamma_{\max}\right].$$

Substituting $\gamma_{\max}$ from (24) completes the proof. $\qquad\square$

# E    The heterogeneous data setting

The analysis in this section is not too different from that before. The main idea here is to refine the upper bound on the distance between the virtual and actual iterates, which can be done using our assumption on bounded data heterogeneity.

**Lemma 10.** *In the setting of Theorem 4, for any $k \geqslant 1$*

$$\mathbb{E}\|\mathbf{x}_k - \hat{\mathbf{x}}_k\|^2 \leqslant 2\mathbb{E}\left[\sum_{m \in [M]\setminus\{m_k\}} \gamma_{\text{next}(k,m)}^2\left[\sigma^2 + 2(M-1)\left(\left\|\nabla F(\mathbf{x}_{\text{prev}(k,m)})\right\|^2 + \zeta^2\right)\right]\right].$$

*Proof.* The argument below is nearly identical to the proof of Lemma 3. We start with an analogue of Lemma 1:

$$\mathbf{x}_k - \hat{\mathbf{x}}_k = \sum_{m \in [M]\setminus\{m_k\}} \gamma_{\text{next}(k,m)}\nabla f_m(\mathbf{x}_{\text{prev}(k,m)}; \xi_{\text{prev}(k,m)}^m),$$

which follows from exactly the same argument. We then use Assumptions 1 and 2 in the same way as in the proof of Lemma 3:

$$
\mathbb{E}\|\mathbf{x}_k - \hat{\mathbf{x}}_k\|^2 = \mathbb{E}\left\|\sum_{m\in[M]\setminus\{m_k\}} \gamma_{\mathrm{next}(k,m)}\nabla f_m(\mathbf{x}_{\mathrm{prev}(k,m)};\xi^m_{\mathrm{prev}(k,m)})\right\|^2
$$

$$
\leqslant 2\mathbb{E}\left[\sigma^2 \sum_{m\in[M]\setminus\{m_k\}} \gamma^2_{\mathrm{next}(k,m)} + \left\|\sum_{m\in[M]\setminus\{m_k\}} \gamma_{\mathrm{next}(k,m)}\nabla F_m(\mathbf{x}_{\mathrm{prev}(k,m)})\right\|^2\right]
$$

$$
\leqslant 2\mathbb{E}\left[\sigma^2 \sum_{m\in[M]\setminus\{m_k\}} \gamma^2_{\mathrm{next}(k,m)} + 2\left\|\sum_{m\in[M]\setminus\{m_k\}} \gamma_{\mathrm{next}(k,m)}\nabla F(\mathbf{x}_{\mathrm{prev}(k,m)})\right\|^2\right.
$$

$$
\left. + 2\left\|\sum_{m\in[M]\setminus\{m_k\}} \gamma_{\mathrm{next}(k,m)}\big(\nabla F_m(\mathbf{x}_{\mathrm{prev}(k,m)}) - \nabla F(\mathbf{x}_{\mathrm{prev}(k,m)})\big)\right\|^2\right]
$$

$$
\leqslant 2\mathbb{E}\left[\sum_{m\in[M]\setminus\{m_k\}} \gamma^2_{\mathrm{next}(k,m)}\Big[\sigma^2 + 2(M-1)\big\|\nabla F(\mathbf{x}_{\mathrm{prev}(k,m)})\big\|^2 + 2(M-1)\zeta^2\Big]\right].
$$

□

**Lemma 11.** *In the setting of Theorem 4, with $\gamma_{\max}$ defined as in (26)*

$$
8L^2 \sum_{k=1}^K \hat{\gamma}_k\|\mathbf{x}_k - \hat{\mathbf{x}}_k\|^2
$$
$$
\leqslant \mathbb{E}\left[\frac{\Delta}{4} + L\gamma_{\max}\sigma^2\left(M\gamma_{\max} + \sum_{k=1}^K \hat{\gamma}_k\right) + \zeta^2\left(2LM^2\gamma^2_{\max} + \frac{1}{2}\sum_{k=1}^K \hat{\gamma}_k\right) + \frac{1}{2}\sum_{k=1}^K \hat{\gamma}_k\|\nabla F(\mathbf{x}_k)\|^2\right].
$$

*Proof.* This follows exactly the same argument as Lemma 8, just replacing that proof's invocation of Lemma 3 with Lemma 10:

$$
\mathbb{E}\left[8L^2 \sum_{k=1}^K \hat{\gamma}_k\|\mathbf{x}_k - \hat{\mathbf{x}}_k\|^2\right]
$$
$$
\leqslant 16L^2\mathbb{E}\left[\sum_{k=1}^K \hat{\gamma}_k \sum_{m\in[M]\setminus\{m_k\}} \gamma^2_{\mathrm{next}(k,m)}\Big[\sigma^2 + 2(M-1)\big(\|\nabla F(\mathbf{x}_{\mathrm{prev}(k,m)})\|^2 + \zeta^2\big)\Big]\right]
$$
$$
\leqslant 16L^2\gamma_{\max}\mathbb{E}\left[(\sigma^2 + 2M(\|\nabla F(\mathbf{x}_0)\|^2 + \zeta^2))\sum_{m=1}^M \gamma^2_{\mathrm{next}(1,m)}\min\{\mathrm{next}(1,m)-1, K\}\right.
$$
$$
\left. + \sum_{k=1}^{K-1}\big(\sigma^2 + 2M(\|\nabla F(\mathbf{x}_k)\|^2) + \zeta^2\big)\hat{\gamma}^2_k(\min\{\mathrm{next}(k+1,m_k)-1, K\} - k)\right].
$$

The stepsize is chosen so that $\gamma_k \leqslant \frac{1}{8L\tau(k)}$, which implies

$$
\gamma_{\mathrm{next}(1,m)}\min\{\mathrm{next}(1,m)-1, K\} \leqslant \frac{1}{8L},
$$
$$
\hat{\gamma}_k(\min\{\mathrm{next}(k+1,m_k)-1, K\} - k) \leqslant \frac{1}{8L}.
$$

Therefore,

$$\mathbb{E}\left[8L^2\sum_{k=1}^{K}\hat{\gamma}_k\|\mathbf{x}_k-\hat{\mathbf{x}}_k\|^2\right]$$

$$\leqslant 2L\gamma_{\max}\mathbb{E}\left[\left(\sigma^2+2M(\|\nabla F(\mathbf{x}_0)\|^2+\zeta^2)\right)\sum_{m=1}^{M}\gamma_{\mathrm{next}(1,m)}\right.$$
$$\left.+\sum_{k=1}^{K-1}\left(\sigma^2+2M(\|\nabla F(\mathbf{x}_k)\|^2)+\zeta^2\right)\hat{\gamma}_k\right]$$

$$\leqslant L\gamma_{\max}\mathbb{E}\left[M\gamma_{\max}\left(\sigma^2+4LM^2\Delta+2M\zeta^2\right)+\sigma^2\sum_{k=1}^{K}\hat{\gamma}_k+2M\sum_{k=1}^{K}\hat{\gamma}_k(\|\nabla F(\mathbf{x}_k)\|^2+\zeta^2)\right].$$

$$\leqslant\mathbb{E}\left[\frac{\Delta}{4}+L\gamma_{\max}\sigma^2\left(M\gamma_{\max}+\sum_{k=1}^{K}\hat{\gamma}_k\right)+\zeta^2\left(2LM^2\gamma_{\max}^2+\frac{1}{2}\sum_{k=1}^{K}\hat{\gamma}_k\right)+\frac{1}{2}\sum_{k=1}^{K}\hat{\gamma}_k\|\nabla F(\mathbf{x}_k)\|^2\right].$$

We used here that $\gamma_k\leqslant 1/(8ML)$ and $\|\nabla F(\mathbf{x}_0)\|^2\leqslant 2L\Delta$. $\qquad\square$

*Proof of Theorem 4.* In this proof, we will use $\gamma_{\max}$ to denote the $\tau(k)$-independent terms in the definition of the stepsize, i.e.,

$$\gamma_{\max}:=\min\left\{\frac{1}{4ML},\sqrt{\frac{\Delta}{KL\sigma^2}}\right\}. \tag{26}$$

As in the data-homogeneous setting, we define the virtual sequence $\hat{\mathbf{x}}_k$ as:

$$\hat{\mathbf{x}}_1:=\mathbf{x}_0-\sum_{m=1}^{M}\gamma_{\mathrm{next}(1,m)}\nabla f_m(\mathbf{x}_0;\xi_0^m)$$
$$\hat{\mathbf{x}}_{k+1}=\hat{\mathbf{x}}_k-\hat{\gamma}_k\nabla f_{m_k}(\mathbf{x}_k;\xi_k^{m_k}), \tag{27}$$
$$\hat{\gamma}_k:=\gamma_{\mathrm{next}(k+1,m_k)}.$$

Denote $\mathbf{g}_k=\nabla f_{m_k}(\mathbf{x}_k;\xi_k^{m_k})$. Using the $L$-smoothness of $F$ and Assumptions 1 and 2, we have:

$$\mathbb{E}[F(\hat{\mathbf{x}}_{k+1})-F(\hat{\mathbf{x}}_k)]$$

$$\leqslant\mathbb{E}\left[-\hat{\gamma}_k\langle\nabla F(\hat{\mathbf{x}}_k),\mathbf{g}_k\rangle+\frac{\hat{\gamma}_k^2 L}{2}\|\mathbf{g}_k\|^2\right]$$

$$\leqslant\mathbb{E}\left[-\hat{\gamma}_k\langle\nabla F(\hat{\mathbf{x}}_k),\nabla F_{m_k}(\mathbf{x}_k)\rangle+\frac{\hat{\gamma}_k^2 L}{2}(\|\nabla F_{m_k}(\mathbf{x}_k)\|^2+\sigma^2)\right]$$

$$\leqslant\mathbb{E}\left[-\frac{\hat{\gamma}_k}{4}\|\nabla F_{m_k}(\mathbf{x}_k)\|^2+\frac{\hat{\gamma}_k}{2}\|\nabla F(\hat{\mathbf{x}}_k)-\nabla F_{m_k}(\mathbf{x}_k)\|^2+\frac{\hat{\gamma}_k^2 L\sigma^2}{2}\right]$$

$$\leqslant\mathbb{E}\left[-\frac{\hat{\gamma}_k}{4}\|\nabla F_{m_k}(\mathbf{x}_k)\|^2+\hat{\gamma}_k\|\nabla F(\hat{\mathbf{x}}_k)-\nabla F(\mathbf{x}_k)\|^2+\hat{\gamma}_k\zeta^2+\frac{\hat{\gamma}_k^2 L\sigma^2}{2}\right]$$

$$\leqslant\mathbb{E}\left[-\frac{\hat{\gamma}_k}{8}\|\nabla F(\mathbf{x}_k)\|^2+L^2\hat{\gamma}_k\|\mathbf{x}_k-\hat{\mathbf{x}}_k\|^2+\frac{5\hat{\gamma}_k\zeta^2}{4}+\frac{\hat{\gamma}_k^2 L\sigma^2}{2}\right].$$

For the third inequality, we used that $\gamma_k\leqslant\frac{1}{2L}$ for all $k$. For the fifth, we used that $\|\nabla F(\mathbf{x})\|^2\leqslant 2\|\nabla F_m(\mathbf{x})\|^2+2\zeta^2$ for any $m$ and $\mathbf{x}$. Rearranging and summing over $k$, we then have

$$\mathbb{E}\left[\sum_{k=1}^{K}\hat{\gamma}_k\|\nabla F(\mathbf{x}_k)\|^2\right]\leqslant\mathbb{E}\left[8(F(\hat{\mathbf{x}}_1)-F^*)+\sum_{k=1}^{K}\left(8L^2\hat{\gamma}_k\|\mathbf{x}_k-\hat{\mathbf{x}}_k\|^2+10\hat{\gamma}_k\zeta^2+4\hat{\gamma}_k^2 L\sigma^2\right)\right].$$

Now, we apply Lemma 11 to bound the second term on the right hand side:

$$\mathbb{E}\left[\sum_{k=1}^{K}\hat{\gamma}_k\|\nabla F(\mathbf{x}_k)\|^2\right]$$

$$\leqslant\mathbb{E}\left[16(F(\hat{\mathbf{x}}_1)-F^*)+\frac{\Delta}{2}+2L\gamma_{\max}\sigma^2\left(M\gamma_{\max}+2\sum_{k=1}^{K}\hat{\gamma}_k\right)+\zeta^2\left(4LM^2\gamma_{\max}^2+11\sum_{k=1}^{K}\hat{\gamma}_k\right)\right].$$

Therefore, if we choose an output $\tilde{\mathbf{x}}_K$ according to

$$\mathbb{P}(\tilde{\mathbf{x}}_K = \mathbf{x}_k) \propto \hat{\gamma}_k \quad \forall k \in [K], \tag{28}$$

then $\mathbb{E}[\|\nabla F(\tilde{\mathbf{x}}_K)\|^2$ is less than this previous expression divided by $\sum_{k=1}^K \hat{\gamma}_k$. By exactly the same argument as for Lemma 9, we can lower bound this sum as:

$$\sum_{k=1}^K \hat{\gamma}_k \geqslant \frac{K\gamma_{\max}}{18}.$$

Therefore, for some constant $c$ (which may change from line to line), we have

$$
\begin{aligned}
&\mathbb{E}[\|\nabla F(\tilde{\mathbf{x}}_K)\|^2 \\
&\leqslant c \cdot \mathbb{E}\left[ \frac{(F(\hat{\mathbf{x}}_1) - F^*)}{K\gamma_{\max}} + \frac{\Delta}{K\gamma_{\max}} + L\gamma_{\max}\sigma^2\left(\frac{M}{K} + 1\right) + \zeta^2\left(\frac{LM^2\gamma_{\max}}{K} + 1\right) \right] \\
&\leqslant c \cdot \mathbb{E}\left[ \frac{(F(\hat{\mathbf{x}}_1) - F^*)}{K\gamma_{\max}} + \frac{\Delta}{K\gamma_{\max}} + L\gamma_{\max}\sigma^2 + \zeta^2 \right].
\end{aligned}
$$

Here, we used that $\gamma_{\max} \leqslant 1/(4ML)$ and $M \leqslant K$. To conclude, we bound

$$
\begin{aligned}
&\mathbb{E}[F(\hat{\mathbf{x}}_1) - F^*] \\
&= \mathbb{E}\left[ F\left( \mathbf{x}_0 - \sum_{m=1}^M \gamma_{\text{next}(1,m)} \nabla f_m(\mathbf{x}_0; \xi_0^m) \right) - F^* \right] \\
&\leqslant \Delta + \mathbb{E}\left[ -\left\langle \nabla F(\mathbf{x}_0), \sum_{m=1}^M \gamma_{\text{next}(1,m)} \nabla F_m(\mathbf{x}_0) \right\rangle + \frac{L}{2}\left\| \sum_{m=1}^M \gamma_{\text{next}(1,m)} \nabla f_m(\mathbf{x}_0; \xi_0^m) \right\|^2 \right] \\
&\leqslant \Delta + \mathbb{E}\left[ \frac{1}{2L}\|\nabla F(\mathbf{x}_0)\|^2 + L\left\| \sum_{m=1}^M \gamma_{\text{next}(1,m)} \nabla F_m(\mathbf{x}_0) \right\|^2 + \frac{L\sigma^2}{2}\sum_{m=1}^M \gamma_{\text{next}(1,m)}^2 \right] \\
&\leqslant 2\Delta + \mathbb{E}\left[ L\left( 2\|\nabla F(\mathbf{x}_0)\|^2 + 2\zeta^2 \right)M\sum_{m=1}^M \gamma_{\text{next}(1,m)}^2 + \frac{ML\sigma^2\gamma_{\max}^2}{2} \right] \\
&\leqslant 2\Delta + \mathbb{E}\left[ LM^2\gamma_{\max}^2\left( 4L\Delta + 2\zeta^2 \right) + \frac{ML\sigma^2\gamma_{\max}^2}{2} \right] \\
&\leqslant 3\Delta + \mathbb{E}\left[ \frac{K\gamma_{\max}\zeta^2}{2} + \frac{KL\sigma^2\gamma_{\max}^2}{2} \right].
\end{aligned}
$$

Plugging this and $\gamma_{\max}$ in above completes the proof. $\qquad\square$