# OpenReview forum: "Asynchronous SGD Beats Minibatch SGD Under Arbitrary Delays"
_NeurIPS.cc/2022/Conference — NeurIPS 2022 Accept_

### Official Review · Reviewer_ZU76 · 2022-07-11

**Rating:** 3
**Confidence:** 3
**Soundness:** 2 fair
**Presentation:** 2 fair
**Contribution:** 2 fair

**Summary:**

Authors propose a new way to analyze convergence of asynchronous SGD that does not depend on the maximum delay of gradient updates among the workers but of the number of machines. This is achieved by introducing delay-adaptive step sizes and a new virtual iterate-based analysis.

**Questions:**

- since the step size is scaled by 1/delay,  if delay is large, would it be just fine to discard those updates alltogether? At least if all machines have data from same distribution.
- in Table 1, what is sigma?
- in Table 1, where is the M/K + sigma/sqrt(K) derived in the paper? It would be good to refer to the theorem and show it in the same form in the paper text as in the table.

**Limitations:**

Limitations not discussed. Societal impact not relevant.

**Strengths And Weaknesses:**

Strengths
- early part of the paper is nicely written, especially the last paragraph of 1.1 illuminated well the problem of previous analysis
- studying asynchronous SGD is important topic, as synchronous SGD has serious scalabiity limitations
- theoretical analysis is structured quite cleanly, and much of the work is in the paper itself instead of appendix
- good language

Weaknesses
- I find it suspicious to compute speedups between two Ohs, i.e  O(a) / O(b) does not make sense, because the underlying constants are not known, and may not be the same for left and right side. This makes me consider the claims of the paper overblown. In addition, to claim async SGD wins minibatch SGD over arbitrary delays seems unrealistic: what if there are very large delays for all gradient communication, i.e arbitrary large delays?  Please rephrase the title.
- there is too much in the paper, so it does not include proper experiments section which would be necessary to validate the (strong) claims.
- paper structure is messy. Hard to see where the results for Table 1 are derived, as they are presented in different form in the theorems.
- paper is missing Conclusion/Summary.

---

> ### Author Response · Authors · 2022-08-01
> **Thank you for your review**
>
> We thank the reviewer for their time. Let us start with the **weaknesses** mentioned by the reviewer:
> >I find it suspicious to compute speedups between two Ohs, i.e O(a) / O(b) does not make sense,
>
> We disagree that comparing big-O complexities to compute the speedups “does not make sense”. This is of course a simplification, but it is a necessary one, since we do not know the precise (tight) constants for the convergence rates. Moreover, our results guarantee that for systems with heterogeneous computation time, Asynchronous SGD is faster than Minibatch SGD **regardless of the constants**.
> >to claim async SGD wins minibatch SGD over arbitrary delays seems unrealistic
>
> We are a bit puzzled by this claim. First of all, Minibatch SGD would suffer from a slow worker even more than Asynchronous SGD as it would halt any progress due to inability to synchronize gradients. Moreover, since we use delay-adaptive stepsizes, an arbitrarily large delay makes the corresponding step very small in magnitude. But most importantly, we do not understand how the reviewer can find a result proven mathematically to be “unrealistic”.
> >experiments section which would be necessary to validate the (strong) claims.
>
> Our claims are not as strong as the reviewer is trying to make them look. Take the example at the end of Section 1.1: If there are two workers and one of them is extremely slow, Minibatch SGD would take just one step and Asynchronous SGD would take $10^6$ steps, one of which will have a huge delay. Is it not apparent why Asynchronous SGD is better in such cases?
> >Hard to see where the results for Table 1 are derived, as they are presented in different form in the theorems.
>
> The part of Table 1 that corresponds to our work (the 3rd row) exactly matches the results in Theorem 3 ignoring constant and logarithmic terms, as stated in the caption.
>
> Answers to the **questions** asked by the reviewer:
> >if delay is large, would it be just fine to discard those updates altogether?
>
> Good question! This approach definitely might work, but it significantly complicates the analysis since we would have to treat workers differently based on whether they are going to make an update or not, and the clean expression for the difference $x_k - \hat{x}_k$ in Lemma 1 would be much messier.
> >in Table 1, what is sigma?
>
>  $\sigma^2$ is the variance of stochastic gradients, defined in the second paragraph of Section 1.5, “Notation and problem setting”
> >in Table 1, where is the M/K + sigma/sqrt(K) derived in the paper? It would be good to refer to the theorem and show it in the same form in the paper text as in the table.
>
> We have two rates like that, given by Theorem 3 in the convex case (see line 228) and non-convex case (see line 233).
>
> Based on these comments, we see that our paper would benefit from adding more details to Table 1. If our submission gets accepted, we will use the extra space in the camera-ready submission to add some additional explanation here.

---

### Official Review · Reviewer_XzpK · 2022-07-12

**Rating:** 7
**Confidence:** 4
**Soundness:** 3 good
**Presentation:** 3 good
**Contribution:** 3 good

**Summary:**

In the paper, the authors proposed a new method to analyze the convergence rate of asyn SGD method. The proposed analysis showed that the convergence rate of asyn SGD is not dependent on the delay of the gradients and comparable or even faster than the convergence of mini-batch SGD under certain conditions.

**Questions:**

1.  Where is the statement "99.9999% of the SGD steps taken have gradients with no delay" obtained?
2. There are a lot of works showing that asynchronous SGD hurt performance, from the theoretical analysis in the paper, asyn SGD should be at least comparable. Could the authors comment on this?

**Strengths And Weaknesses:**

Strenges:
1. Based on the observation that SGD should be robust to the delay and 99.9999% of the SGD steps taken have gradients with no delay, the authors proposed a new analysis for aync-SGD and showed faster convergence rate than previous analysis.

Weaknesses:
1.  Where is the statement "99.9999% of the SGD steps taken have gradients with no delay" obtained?
2. There are a lot of works showing that asynchronous SGD hurt performance, from the theoretical analysis in the paper, asyn SGD should be at least comparable. Could the authors comment on this?

---

> ### Author Response · Authors · 2022-08-01
> **Thank you for your review**
>
> We thank the reviewer for a positive evaluation of our work. Let us respond to the weaknesses/questions provided by the reviewer:
> **Question 1.** The statement that 99.9999% steps have no delay holds only in the specific example that we consider in Section 1.1. In this example, it follows from the fact that the slow worker takes $10^6$ times more seconds to compute an update, so for every delayed step there are $10^6$ steps without any delays. This example was just given for illustrative purposes, and we do not mean to imply that this is the case typically.
> **Question 2**. True, in some practical scenarios, Asynchronous SGD is not the best algorithm, however, it is for reasons other than the convergence rates. The main aspect is that on clusters with homogeneous hardware, such as GPUs of the same kind, the computation time is very well balanced, so there is little need for asynchrony. Since Minibatch SGD supports all-reduce and Asynchronous SGD doesn’t, asynchrony can even hurt the performance due to increased communication time.

---

> > ### Comment · Reviewer_XzpK · 2022-08-09
> > **Reply to authors**
> >
> > Thanks authors for the reply.  The reply to Q2 is not convincing. Existing results show that model trained using asyn SGD are usually worse than model trained using mini-batch SGD. It is not about the training time. However, this paper is still good. I will keep my score.

---

### Official Review · Reviewer_43Xn · 2022-07-12

**Rating:** 8
**Confidence:** 4
**Soundness:** 4 excellent
**Presentation:** 4 excellent
**Contribution:** 4 excellent

**Summary:**

This paper analyzes asynchronous SGD (also known as delayed SGD) where stochastic gradients are computed over $M$ machines and sent to a central server and the server updates its iterates every time it receives a noisy gradient. The challenge in analyzing this algorithm is that some slow machines get to compute the gradients at out-dated iterates.

A baseline for this algorithm is a synchronous version called minibatch SGD where the server waits until all the $M$ machines finish gradient computation and makes a single update out of the aggregated minibatch of $M$ gradients.

Previous analyses on asynchronous SGD assume delays bounded above by $\tau_{max}$ and then show iteration complexities that grow with $\tau_{max}$. This paper analyzes asynchronous SGD with *delay-adaptive step sizes* and proves that asynchronous SGD has the same computational complexity as the baseline minibatch SGD, independent of the delays. This in turn implies that due to the asynchronous nature of SGD (i.e., it doesn't have to wait until all $M$ machines finish gradient calculation), it converges faster than minibatch in wall-clock time.

**Questions:**

Q1. As I detail in L2 below, there is a concurrent work that is very similar to this submission. Can you comment on how your results compare with this concurrent paper?

Q2. In the equation below Line 159, shouldn't the second term have $next(i+1,m_i)$ instead of $next(i,m_i)$, as $next(i,m_i) = i$ always? Correct me if I'm wrong.

**Limitations:**

L1. In Table 1, it is written that the rates for the strongly convex case shown in this paper is $\exp(-\frac{\mu K}{L M}) + \frac{\sigma^2}{K}$. However, from Theorem 3.2 it seems that there should be a leading $M$ factor in the first term. Hiding this leading $M$ term in the strongly convex case is a bad omission in my opinion. (By the way I think $T$ in Theorem 3.2 should be $K$, right?)

L2. This is not necessarily a limitation of this submission, but it is quite unfortunate for the authors that a concurrent paper has presented an overlapping set of results. I am pretty sure that by now the authors would have become aware of the paper "Sharper Convergence Guarantees for Asynchronous SGD for Distributed and Federated Learning" by Koloskova et al. The paper tackles exactly the same problem as this submission and develops a very similar set of theoretical results. Koloskova et al.'s results depend on $\tau_{avg}$ but this can be thought as the number of machines ($M$) in this submission (see their Remark 5). In this regard, Koloskova et al.'s Corollary 7 shows a rate that corresponds to Theorem 2 of this paper, and Corollary 9 of theirs matches with Theorem 3.3 of this submission. Reading the two papers, I came up with a quick list of pros and cons of this paper compared to Koloskova et al.:

Pros:
- Koloskova et al. only cover nonconvex smooth cases.

Cons:
- Koloskova et al. study a slightly more general algorithm which captures both minibatch SGD and asynchronous SGD.
- Koloskova et al. show convergence in heterogeneous case (although with some modification to the algorithm)

…and I'm looking forward to the authors' comments on more detailed comparisons.


**Strengths And Weaknesses:**

S1. This is a solid contribution improving the theoretical analysis of asynchronous SGD. The paper is well-written and conveys the main ideas clearly. Proof sketches are also quite helpful.

S2. Convergence of asynchronous SGD is analyzed in various setups such as Lipschitz convex (Thm 1), smooth Lipschitz nonconvex (Thm 2), smooth convex (Thm 3.1), smooth strongly convex (Thm 3.2), and smooth nonconvex (Thm 3.3) functions.

W1. It is unfortunate that the analysis does not extend well to heterogeneous setups, leaving an extra $\zeta^2$ term that prevents global convergence.

Overall, this paper is a solid theoretical contribution to the community and I would like to recommend acceptance of this paper.

---

> ### Author Response · Authors · 2022-08-01
> **Thank you for the insightful review!**
>
> We thank the reviewer for carefully evaluating our work. We appreciate your positive comments on our contributions. Below we try to address your questions and concerns.
> > **W1**.  It is unfortunate that the analysis does not extend well to heterogeneous setups”.
>
> We do not see this as a limitation of the analysis. The reason that there is an extra $\zeta^2$ term is that the asynchronous SGD method that we consider cannot be guaranteed to converge in situations where the gradients are dissimilar and some workers might not communicate at all. For instance, if the last $M-1$ functions are zero, $F_2=\dotsb=F_{M}=0$ and worker $1$ never returns any gradients, then the convergence could never be better than $\zeta^2$, so our result is exactly what we should expect from the method. A stronger convergence result can be obtained only with extra assumptions on the delays.
>
> **Q1**. We are indeed already familiar with the work of Koloskova et al. It is correct that their analysis covers only the smooth nonconvex setting, while we also considered convex, strongly convex, and nonsmooth. Regarding their results in the heterogeneous case, it is obtained under a very strong Assumption 5, which implies, as far as we can see, that all clients communicate with a fixed frequency. We think that this is an interesting direction for further study of asynchronous algorithms, but their result cannot be compared to ours, since we make no assumptions on the delays. Therefore, despite some overlap in the results, our work gives a different perspective than Koloskova et al., which is neither worse nor better.
> We also got in touch with the authors of the concurrent paper to discuss the overlapping results and potential extensions. In personal communication, they agreed with us that the results in the heterogeneous setting from our works are not directly comparable.
> **Q2**. Thanks for catching this! We should indeed have written $\mathrm{next}(i+1, m_i)$ instead of $\mathrm{next}(i, m_i)$. We checked if the same typo is present anywhere else and it does not seem to be.
> **L1**. Thank you for pointing out the missing factor. In the caption of the table, we mention that we ignore logarithmic terms, and the missing factor of $M$ is logarithmic in the sense that if we want to make the term $M\exp(-\frac{\mu K}{L M})$ smaller than some $\varepsilon$, it is enough to take $K\ge \frac{L M}{\mu}\log\frac{M}{\varepsilon}$. As you can see, it is much more important that $M$ is present inside the exponent than outside. We think it would be best if we clarify in the caption that such factors are also ignored as logarithmic.
> We would like to again express our gratitude for valuable feedback and we hope that our comments shed some light on the paper’s impact.

---

> > ### Comment · Reviewer_43Xn · 2022-08-08
> > **Thanks for the response!**
> >
> > Dear authors,
> >
> > Thanks for your detailed response, this clarifies your contributions quite a lot. Indeed, I agree that the $\zeta^2$ term could be unavoidable unless we make additional assumptions on delays. I also agree that the analysis of heterogeneous settings by your submission and Koloskova et al are interesting in their own right, even though there are some overlapping results in the homogeneous case.
> >
> > As your response clarified most of my concerns and questions, I would like to raise my score to 8.

---

### Meta-Review · Area_Chair_iuGW · 2022-08-23

**Recommendation:** Accept
**Confidence:** Certain

**Metareview:**

The paper makes a significant contribution towards the  analysis of asynchronous SGD, which hinges on a new delay-adaptive step sizes and a new virtual iterate-based analysis. The authors also cover a rather wide range of assumptions, including Lipschitz convex (Thm 1), smooth Lipschitz nonconvex (Thm 2), smooth convex (Thm 3.1), smooth strongly convex (Thm 3.2), and smooth nonconvex (Thm 3.3) functions. All the reviewers understood this contribution, and its novelty. The authors also did a good job reviewing the related literature, and clearly contextualising their work.

**Award:**

No

---

### Decision · Program_Chairs · 2022-09-14

Accept